# Mitigation of Arctic permafrost carbon loss through stratospheric aerosol geoengineering

Yating Chen [1], Aobo Liu [1] & John C. Moore [1,2,3 ✉]

The Arctic is warming far faster than the global average, threatening the release of large amounts of carbon presently stored in frozen permafrost soils. Increasing Earth's albedo by the injection of sulfate aerosols into the stratosphere has been proposed as a way of off-setting some of the adverse effects of climate change. We examine this hypothesis in respect of permafrost carbon-climate feedbacks using the PInc-PanTher process model driven by seven earth system models running the Geoengineering Model Intercomparison Project (GeoMIP) G4 stratospheric aerosol injection scheme to reduce radiative forcing under the Representative Concentration Pathway (RCP) 4.5 scenario. Permafrost carbon released as $CO_2$ is halved and as $CH_4$ by 40% under G4 compared with RCP4.5. Economic losses avoided solely by the roughly 14 Pg carbon kept in permafrost soils amount to about US$ 8.4 trillion by 2070 compared with RCP4.5, and indigenous habits and lifestyles would be better conserved.

[1] College of Global Change and Earth System Science, Beijing Normal University, Beijing 100875, China. [2] CAS Center for Excellence in Tibetan Plateau Earth Sciences, Beijing 100101, China. [3] Arctic Centre, University of Lapland, Rovaniemi, Finland. ✉email: john.moore.bnu@gmail.com

Permafrost covers 24% of the exposed land area in the Northern Hemisphere[1] and at about 1330–1580 Pg[2] contains about twice as much carbon as the atmosphere[3]. The frozen organic content of permafrost is, to a great degree, stabilized by being frozen. Arctic air temperature has increased at a rate of 0.755 °C/decade during 1998–2012, which is more than six times the global average for the same time period[4]. This makes the permafrost both one of the most sensitive ecosystems to warming[5], and also an important future contributor to rising temperatures[6]. Recent observations suggest that every 1 °C increase in temperature will cause the thawing of $4.0^{+1.0}_{-1.1}$ million km$^2$ of permafrost[7]. Model estimates of the permafrost carbon-climate feedback (PCF) factor are 14 to 19 Pg C °C$^{-1}$ under RCP4.5 and RCP8.5 scenarios on decade-to-century timescale[8], and it seems clear from multi Earth System Models (ESMs) that effective mitigation efforts, could substantially reduce the PCF at least over the 21$^{st}$ century[9].

Geoengineering, defined as the deliberate large-scale manipulation of earth's energy balance to mitigate global warming[10], may present a viable proposition for stabilizing carbon storage in circum-Arctic permafrost. Stratospheric aerosol geoengineering (SAI) seeks to mimic the effects of volcanic eruptions that inject sulfur compounds into the lower stratosphere[11]. The warming effect of greenhouse gases is countered by reducing the intensity of solar radiation reaching the surface. Simulation of SAI by ESMs shows that it both mitigates global warming[12] and enhances the terrestrial photosynthesis rate[13] due to increased diffuse solar radiation. But the relative cooling of high latitudes under SAI may offset this increase in photosynthesis[13,14]. Responses of the climate system to SAI are complex[15,16], and many impacts have yet to be simulated. Many have strong reservations about the moral implications of doing, or even researching, SAI if it reduces the motivation to decarbonize and emit less greenhouse gases[10,17,18]. But here we stick to the less controversial assessment of carbon storage in the Arctic permafrost, and the possible economic impact of the PCF. The large-scale permafrost response to geoengineered climates seems to be highly sensitive to the SAI scenario proposed, and the particulars of the ESMs, especially the land surface component used to simulate it[9,14].

In this paper, we investigate the response of soil C stocks to PCF in circum-Arctic permafrost region, leveraging the outputs of ESMs under two climate projections: RCP4.5 and G4. RCP4.5 is arguably the most policy-relevant scenario as the Nationally Determined Contributions (NDCs) greenhouse gas emissions framework would produce similar temperatures trajectories[19], and G4 is one of the most supposedly realistic GeoMIP experiments which uses RCP4.5 as a control run and models stratospheric aerosol injection rather than insolation reduction. Furthermore, the amount of aerosol specified could plausibly be delivered[20], and be relatively effective[21]. The stratospheric sulfate loading required under G4 equates to about ¼ of the 1991 Pinatubo eruption per year, i.e., 5 Tg SO$_2$ per year into the equatorial lower stratosphere beginning in 2020 along with RCP4.5 greenhouse gas emissions[22]. Aerosol injection will end in 2069 under G4, followed by compensatory rapid warming[14]. A more realistic scenario would probably include a ramp up and down of SAI, with various latitudes and altitudes of injection than in the simple G4 specification.

Our results show that the cooling effect of SAI will significantly suppress the temperature rise of permafrost soils, mitigate permafrost carbon-climate feedback, and reduce methane and carbon dioxide emissions from thawing permafrost. Through an integrated assessment model that links the warming potential of PCF with the corresponding economic impacts, we demonstrate that the PCF response to geoengineering proposals is of considerable practical and political as well as ethical and technical interest.

## Results and discussion

**Permafrost C loss**. We note that permafrost C cycle dynamics are poorly represented in CMIP5 models as they were not developed for permafrost soils and do not report vertically resolved soil layers[23]. Instead of directly using the carbon pool or carbon flux outputs of ESMs for analysis, we simulate the large-scale permafrost C response to warming using the PInc-PanTher model[8] modified to include CO$_2$ fertilization effects, forced by the bias-corrected soil temperatures (TSL) and net primary productivity (NPP) simulations of 7 ESMs (Methods; Table 1).

Figure 1 shows the circum-Arctic permafrost C loss simulated by PInc-PanTher when driven by the bias-corrected TSL and NPP of seven ESMs. There is considerable across-model spread, but consistent and significant differences at the 95% level between C losses under the two scenarios (Supplementary Table 1). Between 2020 and 2069, PInc-Panther simulations of soil C change, driven by outputs of 7 ESMs for the RCP4.5 projection, varied from 19.4 Pg C gain to 52.7 Pg C loss (mean 25.6 Pg C loss), while under G4 the ensemble mean was 11.9 Pg C loss (range: 29.2 Pg C gain to 44.9 Pg C loss). Projected C losses are roughly linearly proportional to changes in soil temperature, and each 1 °C warming in the Arctic permafrost would result in ~13.7 Pg C loss; the y-intercept indicates that the Arctic permafrost, if maintained in current state, would remain a weak carbon sink. MIROC-ESM and MIROC-ESM-CHEM, with simulations of warming above 3°C, produce severe soil C losses, while GISS-E2-R with minor soil temperature change produces net soil C gains under both scenarios before 2070. The C losses under the RCP4.5 projection we simulate are close to the range of 12.2–33.4 Pg C (mean 20.8) reported to 2100 by Koven[8], and are within the lower to central part of the range of the 11–135 Pg C reported by Burke[31].

Between 2020 and 2069, warming of the upper 3 m permafrost soils are $1.3 \pm 1.2$ and $2.2 \pm 1.2$ °C under G4 and RCP4.5, respectively, and the ensemble mean PInc-PanTher simulations are net soil C losses throughout the experimental period under both scenarios. Anomalies (G4-RCP4.5) exhibit much less scatter in permafrost C preserved ($-13.7 \pm 4.3$ Pg) due to the well-replicated cooling effect of SAI ($-0.9 \pm 0.4$ °C) among all ESMs that have run G4[32] (Supplementary Fig. 2), indicating that despite model differences, SAI forcing produces consistent permafrost impacts. The difference is significant at the 95% level for 5 of the 7 models (Supplementary Table 1). Thus, the ensemble mean difference G4-RCP4.5 is a halving of soil C released in the circum-Arctic permafrost region with SAI geoengineering. Most simulations of difference between G4 and RCP4.5 scenarios fit well with the linear relationship between $\Delta C$ and $\Delta T$ (Fig. 1), and the CanESM2 and MIROC-ESM-CHEM simulations give the most and least significant difference, respectively.

The time varying NPP we use to drive the PInc-PanTher model is close to steady-state for all the ESMs during the first 5 years (2006–2010) of the simulation (Fig. 1), showing that model drift prior to scenario forcing is negligible. Most ESMs have lower NPP under G4 than RCP4.5 for the permafrost region because the cooling effect suppresses plant photosynthesis, the other models may have higher sensitivity to CO$_2$ fertilization than temperature, or the effects of diffuse light counteract the cooling effects (Supplementary Fig. 3). But the net biological productivity (NBP; defined as NPP minus heterotrophic respiration and ecosystem disturbance) simulations of ESMs show small but positive differences for G4-RCP4.5 (Supplementary Table 2), suggesting that the slowing down of soil heterotrophic respiration due to cooling from SAI is the dominant response in the circum-Arctic permafrost region. The outliers in Fig.1 may be explained by the time-varying soil C input flux related to NPP. For example, HadGEM2-ES which simulates moderate warming and the strongest cooling effects by SAI, produces net soil C gain, due

**Table 1 The models used in this study and their attributes.**

| ESMs (Ref.) | Atmospheric models | Land models | Resolution (lon × lat) [b] | Maximum soil depth (layers) | Dynamic vegetation | Nitrogen cycle |
|---|---|---|---|---|---|---|
| BNU-ESM[24] | CAM3.5 | CoLM | 128 × 64 | 3.6 m (10) | Yes | No |
| CanESM2[25] | AGCM4 | CLASS2.7 | 128 × 64 | 4.1 m (3) | Yes | No |
| HadGEM2-ES[26] | AOGCM | MOSES2.2 | 192 × 145 | 3.0 m (4) | Yes | No |
| GISS-E2-R[27] | GISS-ModelE | Model II-LS | 144 × 90 | 3.3 m (6) | No | No |
| MIROC-ESM[28] | MIROC-AGCM | MATSIRO | 128 × 64 | 14.0 m (6) | Yes | No |
| [a]MIROC-ESM-CHEM[29] | MIROC-AGCM & CHASER | MATSIRO | 128 × 64 | 14.0 m (6) | Yes | No |
| NorESM1-M[30] | CAM4-Oslo | CLM4.0 | 144 × 96 | 42.1 m (15) | No | Yes |

| ESMs (Ref) | Multiple snow layers | Snow insulation effect | Latent heat of soil water | Differing frozen/unfrozen soil thermal conductivity | Snow density ($\rho$) (kg m$^{-3}$) | Snow thermal conductivity (W m$^{-1}$ K$^{-1}$) |
|---|---|---|---|---|---|---|
| BNU-ESM[24] | Yes | Yes | Yes | Yes | f(snow depth; compaction)[c] | $f(\rho^2)$ |
| CanESM2[25] | No | Yes | Yes | Yes | exp($-$time/$\tau$) | $f(\rho^2)$ |
| HadGEM2-ES[26] | No | No | Yes | Yes | Fixed at 250 | Fixed at 0.265 |
| GISS-E2-R[27] | Yes | Yes | Yes | Yes | Fixed at 330 | Fixed at 0.3 |
| MIROC-ESM[28] | Yes | Yes | Yes | No | Fixed at 300 | Fixed at 0.3 |
| [a]MIROC-ESM-CHEM[29] | Yes | Yes | Yes | No | Fixed at 300 | Fixed at 0.3 |
| NorESM1-M[30] | Yes | Yes | Yes | Yes | f(snow depth; compaction)[c] | $f(\rho^2)$ |

The attributes of earth system models (ESMs) listed here are relevant to the soil temperatures (TSL) and net primary productivity (NPP) outputs, which are used to drive a mechanistic soil carbon model for simulating permafrost C dynamics. The microbial decomposition rate of permafrost carbon is a function of soil temperature, and the input flux of permafrost carbon from plant litterfall is assumed to be proportional to NPP at annual scales. Snow schemes and the insulation effects are important for accurate modeling of permafrost soil temperature.
[a]MIROC-ESM-CHEM shares the common features as MIROC-ESM, but with an additional atmospheric-chemistry component (CHASER).
[b]All outputs are downscaled to 0.25 × 0.25° resolution and bias-corrected based on observations and reanalysis data.
[c]Parameterizations that take into account snow depth and compaction process may provide a more accurate simulation.

to the sharp increase in NPP (Supplementary Fig. 3). And the NorESM1-M with a nitrogen cycle but no dynamic vegetation (Table 1) simulates constant NPP throughout the period and gives relatively high soil C loss.

The vertically integrated spatial distribution of permafrost soil C losses in the upper 3 m soil layers for RCP4.5 and G4 scenarios are shown in Fig. 2, and for each model in Supplementary Fig. 4. All seven simulations project much reduced permafrost soil C losses under the G4 experiment and the spatial differences are also consistent. Losses under RCP4.5, and differences under G4 are most pronounced in Siberia, followed by Northern Canada and Alaska, while European Russia and Fennoscandia have least losses and differences. Thus, SAI makes a larger difference at higher latitudes, and with increasing distance from the North Atlantic. This can be understood because SAI tends to move the tropical-pole meridional temperature gradient towards its pre-industrial conditions[33]. Reversing the greenhouse gas forced Arctic amplification effect seen now in warming northern soils[4]. Furthermore, geoengineering mitigates against the greenhouse-gas induced weakening of the Atlantic meridional overturning circulation[34], thus the mild climate conditions produced by the North Atlantic Drift in Northwestern Europe are maintained under SAI, but otherwise weakened under greenhouse gas forcing[35,36]. Moreover, the losses of permafrost C storage are unevenly distributed with depth (Supplementary Fig. 5). The shallow permafrost soils (0–0.3 m) act as a weak carbon sinks under G4 in our simulations, and carbon losses in the upper meter are reduced by ~85% relative to RCP4.5. For the deeper soils (2–3 m), the G4 experiment mitigates about 40% of soil C losses in the circum-Arctic permafrost region.

**Seasonal response.** The exponential relationship between decomposition rate and soil temperature indicates that soil respiration mainly occurs in summer, and therefore a cooling by 1 °C has a greater effect on permafrost carbon preservation if it occurs in a warmer month than a cold one. This implies using

SAI to target permafrost preservation need only be done in the summer season. We calculate the average monthly soil temperature and also the changes between 2020 and 2069 at different soil depths (Fig. 3). The upper meter is where most of the permafrost is predicted to be thawing under warming, and decomposition occurs from June to October when there are also the obvious differences of temperature rise between G4 and RCP4.5 (Fig. 3). At 1–3 m depth, soil temperature rise between 2020 and 2069 is fairly uniform, as is the cooling effect of G4. The summer temperature warming wave is increasingly lagged at deeper soil depths and permafrost thaw occurs as late as January.

**Methane emissions.** Aside from changes in soil C stocks, the other key concern from thawing permafrost is the magnitude of methane ($CH_4$) emissions as they have much larger warming potential per molecule than does $CO_2$[37]. $CH_4$ emission rates depend on many factors, including on degree of waterlogging, in-situ carbon stocks, local warming, soil carbon-nitrogen ratios and biome[38]. There are very few published estimates of the share of methane emissions in the present. It is certainly the case that particular regions will experience increases and other places decreases in wetlands and relative $CH_4$ emissions in the future. But it is also likely, since we are dealing with global averages, that these gains and losses will tend to cancel each other. PInc-PanTher projects the inundated area based on soil C maps, assuming that all Histel (permafrost-affected peat, which covers 1.4 million km$^2$)[39] soils are fully saturated and remain so under warming. We calculate $CH_4$ emissions as a constant scaling factor to relate inundated area and large-scale anoxic respiration rates to $CH_4$ fluxes. This simplified scaling approach is based on expert assessments[2,40] and has been widely used in recent research[41–44]. Methane emissions are presumed to be 1.5%-3.5% of overall soil respiration rates[40,45], and we take a value of 2.3% here. PInc-PanTher simulations of the anoxic respiration rates over the period 2006–2010 are 1.2–1.7 Pg C year$^{-1}$, and so the estimated range of $CH_4$ emissions is 28–39 Tg year$^{-1}$, which is very close to

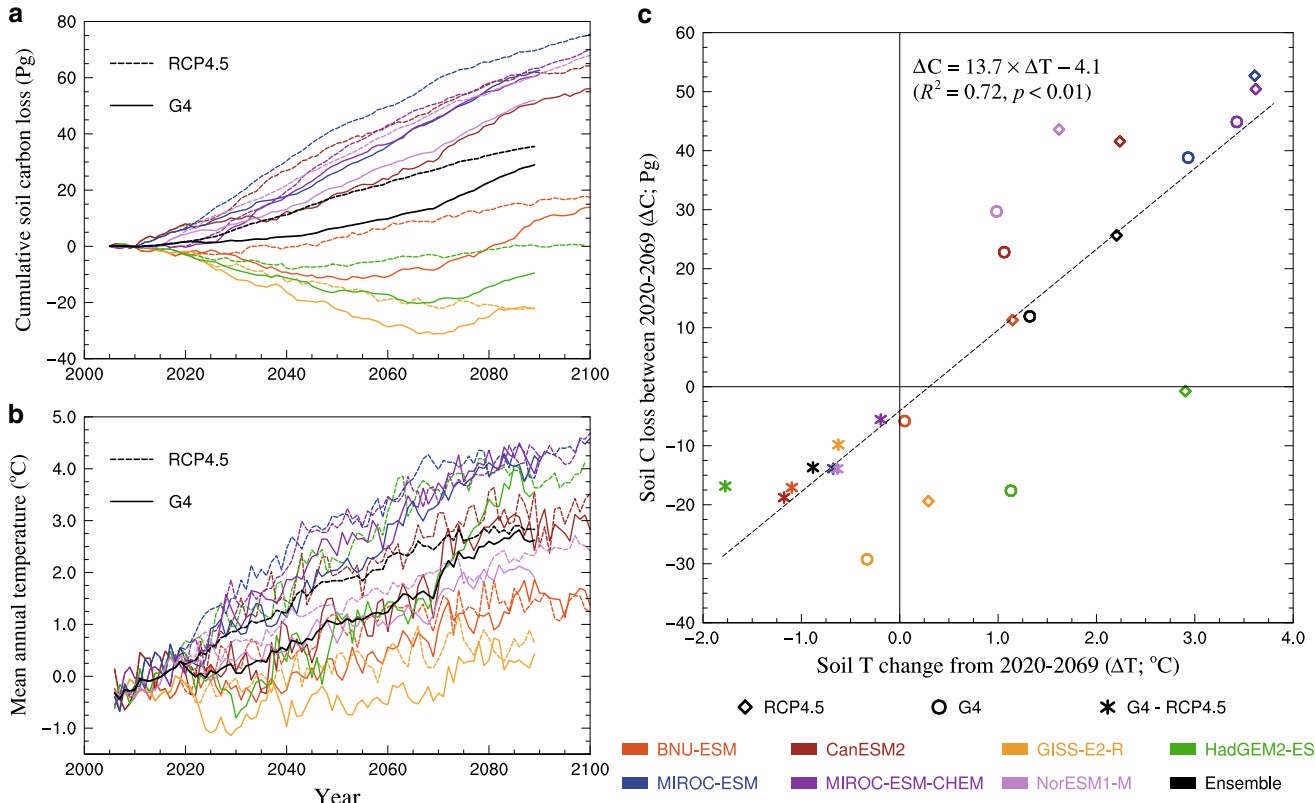

**Fig. 1 Soil temperature change and permafrost carbon loss over time. a** Cumulative losses of circum-Arctic permafrost C storage from PInc-PanTher between 2006 and 2100 under the emission scenario RCP4.5 and the geoengineering experiment G4; each curve is the mean of two methods of calculation (Supplementary Fig. 1; Methods). **b** Mean annual temperature in the upper 3 m of soils, derived from bias-corrected ESM outputs (Table 1). **c** Scatter plot of permafrost C loss and soil T change during SAI implementation (2020-2069); positive values of permafrost C loss indicate losses of soil C stocks to atmosphere. ESMs vary greatly in initial conditions, as they are not tuned for their long-period ocean-atmosphere-sea ice cycles to match the observations on decadal scales. Therefore, we perform the initial bias-correction and utilize ensemble means.

the 15–40 Tg $CH_4$ year$^{-1}$ estimates of current permafrost wetland $CH_4$ emissions[46]. Since climate under G4 would be drier than under RCP4.5[36,47], assuming the same proportionate release of $CH_4$ under both scenarios might tend to minimize (RCP4.5-G4) differences, and so be regarded as a conservative estimate.

Figure 4 shows the changes of permafrost $CH_4$ emissions, due to PCF under the RCP4.5 and G4 scenarios. Each trajectory is normalized by subtracting the 2006–2010 mean emissions. Between 2020 and 2069 under G4, methane emissions are predicted to be about 40% lower than under RCP4.5, a difference significant at the 95% level for all ESMs (Supplementary Table 1). This ratio indicates that under the G4 scenario, the increase in soil respiration caused by PCF will be significantly suppressed by SAI.

**Sources of uncertainty in C estimates**. The ESMs differ in very many ways and Table 1 lists only a few of them. Moore et al.[36] found that 4 of the same models as used here also produce a similar spread of differences (RCP4.5-G4) in mass balance for Greenland as found for C release from permafrost. This results from combinations of reasons: differences in the sensitivities of ESMs to greenhouse gas and SAI forcing means that simulated changes in AMOC, seasonal sea ice, cloud cover, specific humidity and hence longwave radiative absorption and accumulation rates all differ. A recent analysis[48] of ESM differences in the Arctic across 13 CMIP5 models including BNU-ESM, MIROC, NorESM1-M, CanESM2, GISS-E2-R, explains it largely in terms of differing estimates of surface albedo and Planck feedbacks which show the largest intermodel differences. Their analysis indicates

that these differences arise not only from different degrees of simulated Arctic warming but also are partly related to the large differences in initial sea ice cover and surface temperatures[48]. Supplementary Fig. 6 shows the differences (G4-RCP4.5) between soil C annual losses as a function of differences in TSL and surface temperatures. There are marked differences between models, but it is clear that TSL is a far better predictor of C losses than surface temperature forcing, with TSL typically accounting for 30% of variance in C, whereas surface T reflects only 10% of the variability, and is not a significant predictor for 5 of the 7 ESMs (Supplementary Fig. 6). Even though MIROC-ESM and MIROC-ESM-CHEM share the same land surface module (Table 1), TSL accounts for about 20% of variability in C loss for MIROC-ESM, but twice as much with MIROC-ESM-CHEM. This spans almost the full range of sensitivity of C losses to changes in TSL. The remarkable differences in TSL change (G4-RCP4.5) between the two MIROC models may also be seen by comparing the monthly TSL difference (RCP4.5-G4) for all the ESMs (Supplementary Fig. 7). This suggests that natural variability over the 50-year simulations is as large as differences due to model formulation, and hence supports the Block et al.[48] view that initial conditions for Arctic sea ice and phases of the long period climate cycles driving temperature play very important roles in explaining across ESM variability, and strengthens the utility of the multi-model ensemble mean as a way of neutralizing these.

In the case of permafrost thermal state, the lowest soil boundary is a critical uncertainty affecting the simulation of permafrost[49]. Earlier generations of land surface model typically had a lower boundary at 3 m soil depth, but in general the deeper

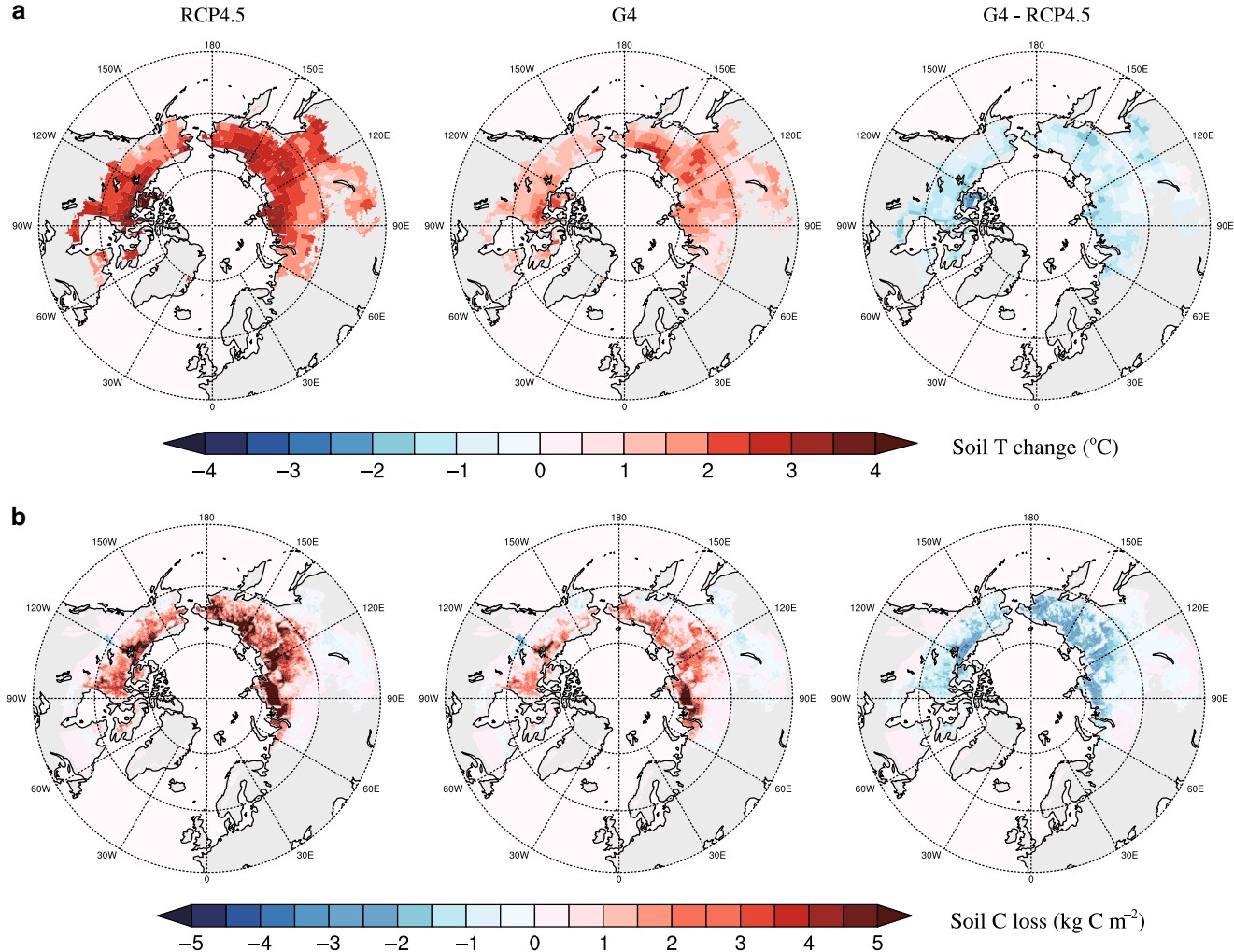

**Fig. 2 The cooling effect of geoengineering and projected soil carbon loss.** Ensemble mean spatial distributions of (**a**) permafrost soil temperature change in the upper 3 m of soils between 2020 and 2069 under the emission scenario RCP4.5 and the geoengineering experiment G4, as well as the difference (G4-RCP4.5), derived from the TSL outputs of 7 ESMs after bias-correction and (**b**) permafrost soil C loss from PInc-PanTher between 2020 and 2069, integrated from surface to 3 m depth. By combining the compiled soil carbon maps with laboratory incubation syntheses (Methods), PInc-PanTher provides data-constrained estimates on the permafrost C response to imposed climatic warming.

the soil layers the better the model is at simulating the thermal condition. Some models have substantial biases in their soil temperature simulations (Supplementary Fig. 8), that have been attributed to the way snow cover, vegetation and soil properties are parameterized[50]. Soil thermodynamics, and hence near-surface permafrost extent depend greatly on the accuracy of the simulated snow cover[51]. Additionally, simulated surface vegetation, soil organic matter and hydrology impact the surface energy balance and help determine model permafrost distribution[52]. Wang et al.[53] reviewed the performance of various land surface models: those with that include vertical structure via multilayer snow schemes simulate stronger and more realistic insulation, better capturing the observed nonlinear profiles of snow temperature with depth[54]. Incorporating wet-snow metamorphism accounts for stored and refrozen liquid water and allows for better parameterization of snow compaction and snow thermal conductivity (Table 1), improving snow depth and surface soil temperatures[53]. These factors all interact and obscure relationships, and we find that soil C losses did not exhibit statistically significant relationships with snow cover in the ESMs.

In comparison with the land surface models, differences in the treatment of stratospheric aerosol physics and chemistry[47]

probably plays a minor role. For example, HadGEM2-ES adopts a stratospheric aerosol schemes to simulate the sulfate aerosol optical depth (AOD). BNU-ESM and MIROC-ESM use the prescribed meridional distribution of AOD recommended by the GeoMIP protocol[55]. MIROC-ESM and MIROC-ESM-CHEM share most processes except for an interactive atmospheric chemistry module in MIROC-ESM-CHEM[56]. But comparing the BNU-ESM results in Fig.1 with MIROC-ESM and MIROC-ESM-CHEM (which share similar SAI schemes), we see that BNU-ESM produces much lower soil C losses than the other two which have very similar results. This is likely because BNU and both MIROC models have different land surface models, soil depths and different ways of treating the snow cover (Table 1). For the purposes of permafrost stability the general pattern of cooling produced seems dominant as soil temperatures appear the single most critical variable in determining the differences in response between G4 and RCP4.5. The patterns of cooling are well-replicated by all ESMs that have run G4[32] (Supplementary Fig. 2).

The lifespans of stratospheric sulfur aerosols in different regions and altitudes vary from a few months to several years[57]. Since the G4 experiment does not remove $CO_2$ from the atmosphere, if radiative forcing by SAI were to suddenly cease,

**Fig. 3 Monthly soil temperature and its changes between 2020 and 2069.** Histograms with error bars (left axes) show the monthly soil temperature change over the period 2020-2069 under the emission scenario RCP4.5 and the geoengineering experiment G4, and line plots (right axes) show the mean soil temperatures at different soil depths (**a**) 0–30 cm, (**b**) 30–100 cm, (**c**) 100–200 cm and (**d**) 200–300 cm over the same period. Soil temperatures are ensemble means from the bias-corrected ESM outputs. The cooling effect of SAI is significant from May to October, mitigating summer warming and retaining carbon from permafrost.

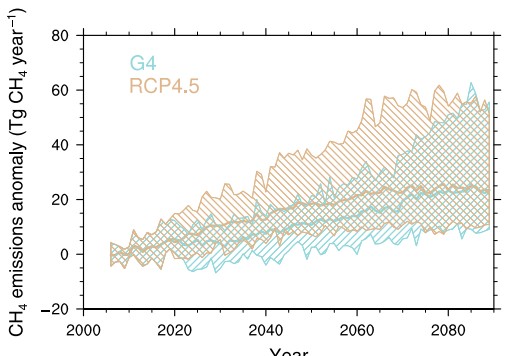

**Fig. 4 Changes in permafrost methane emissions.** Projected annual methane ($CH_4$) emissions relative to the first five years (2006–2010) of the simulations under the emission scenario RCP4.5 and the geoengineering experiment G4. Thick lines show ensemble mean values and hatched areas show the across-ESMs range. The $CH_4$ fluxes in the permafrost region are inferred from the anoxic respiration rates in the inundated area.

there would be a compensatory rapid global temperature rise, the termination shock[58]. The rise in surface air temperatures following termination causes a strong air-soil temperature gradient that drives heat into the soil generating a sudden

increase in permafrost emissions from 2070-2089 which partially offsets the carbon loses avoided during the G4 implementation (Supplementary Fig. 9 and Supplementary Table 1). Hopefully, such an unplanned termination strategy would not be employed in reality[59].

**Economic implications of permafrost loss.** It is estimated that the permafrost area would eventually be reduced by over 40% if climate is stabilized at 2 °C above pre-industrial levels[7]. We can explore the impact of G4 on future permafrost extent as well as active layer thickness distributions by using one definition of permafrost extent based on mean annual ground temperature (MAGT), and the NorESM1-M model which has deepest soil depth of the seven in Table 1 (an important factor in successful permafrost diagnosis[50]). In our simulations (Supplementary Fig. 10), the permafrost area would eventually be reduced by 15% under G4 and 35% under RCP4.5.

The remaining global carbon budget for reaching the 2 °C warming target is about 400 Pg C[60,61], but the emitted carbon from thawing permafrost could reduce it by about 30 Pg C[61,62]. Thus, the 6–19 Pg C (mean 14) not released by permafrost under G4 compared with RCP4.5 is a small, but significant fraction of the remaining emission budget. By noting that $CH_4$ warms 12.4 times as effectively as $CO_2$ per unit soil C lost over a 100-year

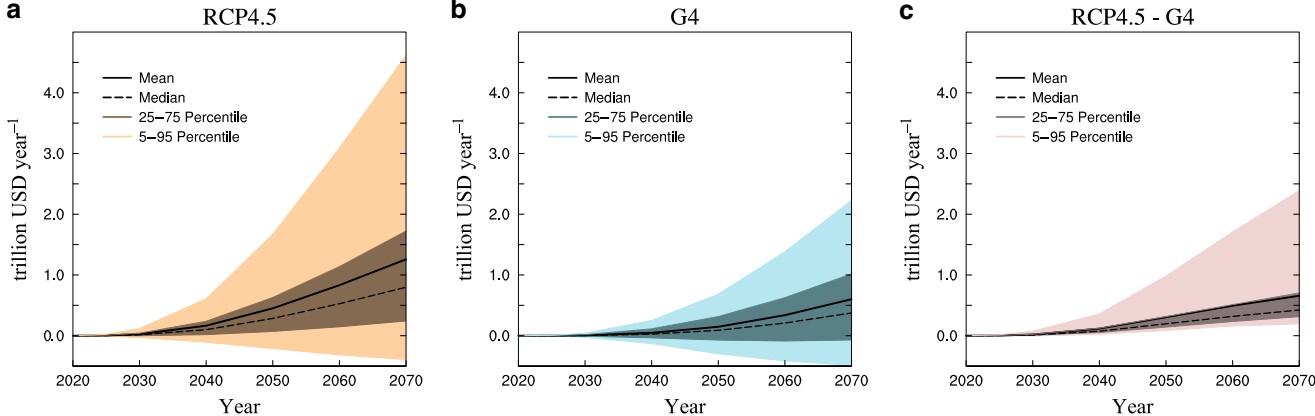

**Fig. 5 Economic impacts of permafrost CO₂ and CH₄ emissions.** Predicted annual economic losses due to PCF under (**a**) the geoengineering experiment G4 and (**b**) the emission scenario RCP4.5, as well as (**c**) the difference (RCP4.5–G4), obtained from 100,000 Monte-Carlo runs of PAGE-ICE driven by projected permafrost CO₂ and CH₄ emissions. Solid black lines represent the mean values; dotted black lines show the median values; light and dark color intervals show 5–95% and 25–75% confidence intervals, respectively. Negative values under RCP4.5 and G4 indicate that the PCF may bring economic benefits due to the net C gains simulated by GISS-E2-R and HadGEM2-ES.

time-frame[63], the differences in methane emissions between G4 and RCP4.5 give an additional 26% reduction in radiative forcing beyond the CO₂ emissions compared with the RCP4.5 scenario between 2020 and 2069.

Moreover, the release of CO₂ and CH₄ into the atmosphere will result in socio-economic impacts in the Arctic due to, for example, infrastructure degradation in thawing ground, loss of indigenous villages and ways of life, changes in the ecology of the tundra and livestock. Hope and Schaefer[64] suggested that under the medium emission scenario (A1B) and aggressive abatement policy (2015r5low), the total economic impacts of carbon emissions from thawing permafrost by 2200 can reach US$ 43 trillion and 20 trillion, respectively. A recent study estimated that the changing Arctic would increase economic impacts of climate change by $33.8 trillion for 2.0 °C target, and $66.9 trillion for the Paris NDC commitments[65]. These estimates were aggregated until 2300 and derived from the PAGE-ICE integrated assessment model[65], which primarily considers two feedbacks in the Arctic: permafrost carbon feedback (PCF) and surface albedo feedback. An estimate of the relative contribution of avoided PCF to total avoided warming by SAI might be helpful. However, the analysis of the SAI economy should include not only on the positive benefits of avoiding warming, but also on the possible negative effects of SAI implementation. The economic advantages of SAI are simply enormous, but the negative effects are very difficult to calculate because of the unknown unknowns - that is unexpected surprises. Bickel and Agrawal[66] suggested that even considering the probability and side effects of aborting geoengineering, SAI may still pass a cost-benefit test over a wide range of scenarios. Another approach has been to look at economic disparities under SAI compared with pure greenhouse gas forcing[67], although no analysis was done for G4 or RCP4.5, disparities are greatly reduced with SAI which is hence beneficial to social stability. Although analyzing the cost-benefit of SAI as a whole, is fraught with difficulty because of unforeseen and presently unquantifiable damages from SAI, here we used the PAGE-ICE model to evaluate the global economic impacts of PCF, and for reference total economic damage as well. We drive the PAGE-ICE model with permafrost CO₂ and CH₄ emissions under G4 and RCP4.5 scenarios (Methods). Under the Paris NDC commitments, the emission pathway is likely to resemble RCP4.5[19] and the socio-economic scenario be similar to IPCC SSP2.

Systematic uncertainties in the projected annual economic losses due to PCF are explored with uncertainty (mean ± 1σ,

assuming a Gaussian distribution) from permafrost emissions and a 100,000 member ensemble suite of simulations in Fig. 5, (and the differences between RCP4.5 and G4 scenarios for each model in Supplementary Fig. 11). Net economic losses reduced by SAI are most for the CanESM2 simulations and least for MIROC-ESM-CHEM (Supplementary Fig. 11), consistent with the ESMs estimates of mitigating permafrost C loss. The simulated annual economic losses and gains are very uncertain, with even about ¼ of the ensemble suggesting net economic benefits from the PCF under RCP4.5 and G4. This is due to GISS-E2-R and HadGEM2-ES simulating net C gains under the scenarios. The RCP4.5-G4 5–95% range differences are all positive with mean of US$ 0.6 trillion/yr and 5–95% range of US$ 0.2–2.4 trillion/yr in 2069 (Fig. 5). With consideration of the time value of money and losses in utility, we estimated the cumulative sum of discounted economic impacts during 2020-2069. We expect that the PCF will generate estimated net economic losses of US$ 13.8 trillion (5–95% range: US$ −5.4 to 51.2 trillion) under RCP4.5 and US$ 5.4 trillion (5–95% range: US$ −8.0 to 22.2 trillion) under G4. Hence G4 would reduce economic losses by ~60% (US$ 8.4 trillion) compared with RCP4.5. Our estimates of PCF economic impacts for G4 and RCP4.5 are quite similar to those for the 2.0 °C target and NDCs from the Paris Agreement[65], but somewhat lower mainly because our period of concern is the shorter (2020–2069) period rather than through to 2300. For total economic loss estimates we used the PAGE-ICE model to assess the economic impacts of climate change, and found that under the RCP4.5, 2.5 and 2 °C targets[65], the economic losses due to climate change in 2069 will be US$ 22, 10 and 6 trillion per year, respectively (Supplementary Fig. 12). We consider these estimates simply as a comparative guide to see the PCF in context, and so crudely, we may consider US$12–16 trillion per year as the economic benefit brought by G4's avoiding warming in 2069. The economic benefits by avoided PCF in 2069 would be about US$ 0.6 trillion per year, thus the relative contribution of avoided PCF to total avoided warming by SAI would be about 1/20 to 1/25 of the total.

The implementation costs of the G4 scenario are estimated to range from about US$ 0.05 trillion to US$ 0.4 trillion[20], in other words the G4 type of SAI might well cost less than the savings due only to PCF. The economic impacts of PCF are most pronounced in the non-market sector (e.g., ecosystem damages), rather than market sectors (e.g., forestry, tourism etc.) and sea level rise (i.e., coastal flooding), and these calculations do not include Arctic

infrastructure and building damage caused by thawing permafrost. Subsea permafrost is not considered in this study, but geoengineering would also mitigate the methane emissions from subsea permafrost degradation by directly affecting the ocean temperatures[34] and by indirectly affecting the extent of sea ice[68] which serves as a natural physical barrier[69]. We also do not consider possible impacts of changing marginal permafrost zones over time. Thus, we implicitly assume that the large amounts of C released from thermokarst morphological features such as melt lakes[70], will remain the same as at present and in the PInc-PanTher calibration.

Despite model uncertainties this study indicates SAI, particularly if implemented during the summer period over the Arctic, can halve permafrost C losses compared with RCP4.5 and $CH_4$ emissions by 40%. The analyses undertaken here are far from the last word in sophisticated permafrost carbon modeling, but they give clear directions for future research. SAI and its impacts should be studied in more depth, especially using models tailored for the local implications on human health, droughts, flooding, extreme events and agriculture. This is especially important in the developing world where climate impacts are already being strongly felt.

## Methods

**The modified PInc-PanTher model.** PInc-PanTher is a data-based model developed by Koven et al[8]. It is based on a syntheses of laboratory incubation work on plant decomposition that are highly constrained by data specifically from permafrost soils. The standard PInc-PanTher model ignores changes in vegetative productivity due to climate change and hence may overestimate the soil C release[9]. In view of this, we simulate the response of plant productivity to climate warming and $CO_2$ fertilization by introducing a time-varying input flux rather than a fixed one. We assume an initial steady state to infer the initial carbon fluxes into the soil pools. Since NPP increases with greenhouse gas concentration, we assume an increase in the input flux proportional to NPP and scale it between layers according to the initial state. According to a previous study[8], large-scale thawing of permafrost deeper than 3 meters will not occur under the RCP4.5 scenario, and the majority (90%) of calculated emissions come from surface soils less than 3 m deep. Since G4 reduces soil temperature relative to RCP4.5, we only focus on surface soils. We consider permafrost degradation processes due to the warming of soils, the lengthening of thaw period and the deepening of the active layer.

Soil C maps covering the Arctic permafrost region (NCSCDv2)[39] were regridded using a mass-conservative interpolation to $0.25 \times 0.25°$ spatial resolution, then regridded vertically to four layers: 0–0.3, 0–1, 1–2, and 2–3 m. This process facilitates comparison and synthesis, while the mass-conservative interpolation method ensures that regridding effects are within ±0.5%. We take an initial permafrost soil C storage of 727 Pg C in the upper 3 m of the Arctic permafrost. The soil carbon pools in our modified model are the means from two methods (Supplementary Fig. 1): dependent on soil C:N ratio[71] and dependent on the ratio of mineral and organic material in the soil[72]. We examined many ESMs fields such as snow cover, precipitation etc., but find that only soil temperature (TSL) and net primary productivity (NPP) have appreciable predictive power. PInc-PanTher responds to changes in snow depth and insulation properties affect soil temperature profiles. The ESMs used here have good snow insulation models[53] and the systematic errors in soil temperature introduced by ESMs errors in snow distribution[53], should be eliminated by the bias correction procedure in soil temperatures (Supplementary Fig. S8).

Permafrost carbon cycle dynamics relate to three factors: metabolic processes of microbial decomposition, soil thermodynamics, and input C flux from plant litterfall. To express the response of vegetation production to climate change, we assume that the soil C input flux is proportional to NPP at annual scales in every layer. We set the initial input flux to satisfy initial conditions that soil C losses balance inputs for the period 2006–2010. The permafrost C stock ($C_p$) varies over time as:

$$\frac{dC_p}{dt} = P(t) - C_p \times k \times Q_{10}(tsl) \qquad (1)$$

where $P(t)$ is the litterfall input flux that increases over time; $k$ is the decay constant at the reference temperature (5°C); and $Q_{10}(tsl)$ is a function of soil temperature that controls the decomposition rates. The standard PInc-PanTher uses a truncated $Q_{10}$ function, which assumes zero soil respiration when soil temperature is at, or below, the freezing point[8]. However, decomposition in winter does proceed in the upper layer of soils, so we assume that the winter respiration rates of the upper layer increase by a factor of 2.9 per 10 °C soil temperature rise[73].

**Bias correction and downscaling.** We drive the modified PInc-PanTher with all ESMs that have the monthly mean values of soil temperature and annual NPP, available for G4, RCP4.5 as well as the historical period (Table 1). The ESMs were bi-linearly interpolated into the same $0.25° \times 0.25°$ grid. ESMs produce a wide range of soil temperatures and NPP, so we bias-corrected the downscaled soil temperatures with the trend-preserving, Inter-Sectoral Impact Model Inter-comparison Project (ISI-MIP[74]) method using 458 Russian meteorological stations (2006-2015) for Siberia and daily data from RIHMI-WDC[75] and ERA5[76] reanalysis data from 2006-2015 for the rest of the region (Supplementary Fig. 8). The same methods were used to correct ESMs NPP with the continuous and widely used MODIS annual NPP products[77] (Supplementary Fig. 3). The soil temperatures of MIROC-ESM, MIROC-ESM-CHEM and NorESM1-M had least bias, and also deepest soil layers (Table 1).

**PAGE-ICE integrated assessment model.** PAGE-ICE[65] is the latest version of PAGE (Policy Analysis of Greenhouse Effect[64]) model, which has been widely used for policy assessments, and was one of three models adopted by the U.S. government to estimate the social cost of carbon emissions[78]. The E.U. used the PAGE-ICE model to evaluate the additional cost associated with Arctic change[65]. Due to advances in climate and socio-economic sciences, such as adjusted climate response parameters based on CMIP5 models, revised $CO_2$ cycle equations and new economic impact functions, PAGE-ICE provides a more accurate assessment of future climate change and related economic impacts compared with PAGE.

Here we added the permafrost annual $CO_2$ and $CH_4$ emissions we derived, to the RCP4.5 anthropogenic emissions projections which follow the SSP2 socio-economic scenario. Hence deriving estimates of the economic losses caused solely by PCF through the PAGE-ICE model. PAGE-ICE uses a radiative balance climate model to simulate global and regional temperature changes and sea-level rise. Estimates of the climate change damages to society over time are based on damage functions that relate GDP losses to global and regional mean temperature change and sea-level change. Warming and sea-level rise cause damage in 4 different sectors[79]: market sectors (e.g., agriculture, forestry, tourism etc.), non-market sectors (e.g., mortality and ecosystem damages), sea-level rise (i.e., coastal flooding) and a stochastic discontinuity (a one-time tipping point resulting in a large loss of GDP).

PAGE-ICE provides six alternative damage functions based in part on macroeconomic analysis of the effect of historical temperature shocks on economic growth[80]. In this study, we choose the PAGE09 & IPCC AR5 & Burke parameterization scheme, in which a statistical fitting was used to meet the IPCC AR5 and Burke global impact estimates. PAGE-ICE models an exogenously-determined adaptation policy that reduces impacts in three of the sectors for a price. The costs of reducing emissions below their business-as-usual path (called preventative costs in PAGE-ICE) are also calculated. Finally, all impacts are then equity weighted, discounted at the consumption rate of interest[81] and summed over the whole period. All results reported are simulated for 100,000 runs to perturb various model parameters and fully explore the economic impacts. Parametric uncertainties are designed with triangular distributions, and all calculations are done in a probabilistic way using Latin hypercube sampling to establish the probability distribution of the result.

## Data availability

ESMs data can be downloaded from ESGF-Node (https://esgf-node.llnl.gov/search/esgf-llnl); MODIS NPP products from NASA (https://modis.gsfc.nasa.gov/data/dataprod); Russian meteorological data from RIHMI-WDC (http://meteo.ru/english/climate/soil.php); ERA5 reanalysis data from ECMWF (https://cds.climate.copernicus.eu); Additional data that support the findings of this study are available from the authors upon reasonable request. The source data underlying Figs. 1 and 3–5 are provided as a Source Data file.

## Code availability

PAGE-ICE software can be downloaded from ref. [65] standard PInc-PanTher model from ref. [8] bias correction ISI-MIP method from [https://github.com/SantanderMetGroup/downscaleR]; the data analysis and figure drawing computer codes are in NCAR Command Language (NCL) scripts and are available upon request.

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

## Acknowledgements

We thank all participants of the GeoMIP and their model development teams; the CLIVAR/WCRP Working Group on Coupled Modeling for endorsing the GeoMIP; and the scientists managing the earth system grid data nodes who have assisted with making GeoMIP output available. This research was funded by the National Basic Research Program of China (2016YFA0602701).

## Author contributions

Y.C. and A.L. performed the experiments and analyzed the results. J.C.M. and Y.C. designed the experiments and interpreted the results. All authors contributed to preparation of manuscript text.

## Competing interests

The authors declare no competing interests.
