## [Peer Review File · Nature Communications]

Reviewers' comments:

Reviewer #1 (Remarks to the Author):

This study addresses whether injection of sulfate aerosols into the stratosphere can mitigate permafrost carbon climate feedback. The authors use the PInc-PanTher model of permafrost carbon-climate feedbacks driven by seven earth system models running the Geoengineering Model Intercomparison Project (GeoMIP) G4 stratospheric aerosol injection scheme to reduce radiative forcing under the Representative Concentration Pathway (RCP) 4.5 scenario. They find that permafrost carbon released as CO₂ is halved and as CH₄ by 40% under G4 compared with RCP4.5. They further assess that by mitigating ~13 Pg C in permafrost soils leads to about US\$8.0 trillion in economic value compared with RCP4.5.

The idea behind this study is very interesting and novel, where the authors convert potential carbon 'not lost' as a result of climate mitigation into economic value. This can be very useful in the policy point of view when discussing the application of geoengineering. And since SAI application is actually being discussed to be deployed as a pilot study, this kind of analysis is quite timely.

The general consensus of land surface models used in the CMIP5 experiments is that these models do not represent permafrost carbon very well. There are two main reasons for this: first, the models do not represent permafrost and vertically resolved soil layers and second, the models do not represent the complexity of carbon cycling. Due to this caveat, the authors chose to use PInc-PanTher model driven by GeoMIP G4 as well as RCP4.5 scenario to quantify differences in CO₂ and CH₄ release. Although the general idea behind this study is interesting, I have concerns about the methods used in this study and the validity of using PInc-PanTher model for this analysis.

First, permafrost carbon release greatly depends on temperature, particularly soil temperature. Therefore, the results in permafrost C losses depend solely on how different models exhibit soil temperatures under different scenarios. However, the authors only focus on permafrost soil C. There is a wider variety of range in soil temperatures and specifically different soil temperatures at depth. The results should focus more on why this is different. I don't see why getting the Ensemble mean would be important and interesting in this case without this discussion. Permafrost soil temperatures at different depths highly depend on how the models represent permafrost. But the authors fail to address and discuss this important point. Second, estimating permafrost C loss in CH₄ is an important aspect of this study. The tricky part of accurately projecting CH₄ emissions under permafrost thaw relies heavily on the projections of inundated area created under permafrost thaw. In PInc-PanTher, CH₄ emissions are a constant fraction of heterotrophic respiration from anoxic soils. Therefore, the CH₄ release from permafrost C estimated in this study seems a very crude estimation. Third, as PInc-PanTher is a simple model that simulates soil carbon, the authors use NPP and NBP from CMIP5 models. Although, I wonder whether using NBP is a good choice in this case. NBP, by definition includes heterotrophic respiration in the term itself. If the authors are concerned about validity of using CMIP5 models in their soil C (and they are), they should not use this term. In addition, NBP includes ecosystem disturbance term such as fire, which in most of these models do not have interactive fire model, NBP is not a good choice in parameter comparison.

Below are some of specific questions and comments I have throughout the manuscript.

- L12: It is unclear what the hypothesis the authors are testing. Either clarify or revise this sentence.
- L41-42: To me, the motivation to make the comparison between G4 and RCP4.5 scenarios is unclear. The readers will benefit from some explanation.
- L46: The authors should give reasoning why they are using PInc-PanTher model rather than permafrost soil C release directly from ESMs.

- L51: Is this circum-Arctic permafrost?
- Section 'Permafrost C loss': What about the difference in soil T between G4 and RCP4.5? What is the magnitude? Also, why didn't the authors compare the difference between RCP8.5+Geoengineering with RCP4.5 where the temperature levels are similar? The authors should show temperature-time series plot and discuss why there is such large variability in permafrost C loss across the models.
- L53-55: This is interesting. How? When you look at the difference in global C cycling (or even soil C) across the CMIP5 models, there is a very large spread. Does this mean that soil T is not playing much effect on this? Can the authors go into this a bit in more detail? Again, showing the soil T of all the models would be helpful.
- L61-62: I am very confused about this statement because only two out of the seven models show lower NPP under G4 compared to RCP4.5 (Fig.s2). I suspect that the other models have higher sensitivity to CO₂ fertilization over temperature. This needs to be pointed out.
- L63-64: I don't understand this transition. Please consider revising.
- L70-72: This is understandable because first, vegetation response is quite constrained to temperature in high latitude ecosystems and the growing season is very short. In other regions, this may not be the case as vegetation response can be due to multiple factors such as precipitation, CO₂ fertilization, and radiation. But I do not understand the authors' reasoning why this is important. Is it due to the C input to soil? Please explain.
- L107-108: Q10 is temperature sensitivity of decomposition, not decomposition rate itself. In this case, the authors may be better off directly showing the rate of decomposition (k) or heterotrophic respiration than Q10. To me, the two lines look very similar and without any statistical analysis, it could be an overstatement of the authors to say that the two simulations exhibit different soil temperatures.
- L112-114: Could the authors emphasize here that because SAI is a radiation management method it is expected that it is not so effective in the winter months where the length of solar period is shorter?
- L118: Why do the authors assume decomposition does not occur below 0C? Is this because of how PInc-PanTher operates?
- Section 'Uncertainties in C estimates': The authors fail to recognize or discuss the differences in how different models simulate physical processes associated to permafrost. This has a large impact on soil temperature, particularly deep soil temperature. As a result, the difference in permafrost recognition across models are really causing the large variability. This is a very important aspect, so should be discussed here.
- Throughout: In this version of the manuscript, the way it is written can make the reader very confused about what the authors did. It often sounds like the authors are actually using the CMIP5 simulations, but actually what the authors did was to use soil T data extracted from the CMIP5 models to force PInc-PanTher model. This needs to be better portrayed in the writing.

Reviewer #2 (Remarks to the Author):

The paper by Chen et al. addresses the impact of stratospheric geoengineering on mitigation of carbon and methane loss from Arctic permafrost. Stratospheric geoengineering was obtained from the GeoMIP G4 stratospheric injection scheme scenario (injection approximately 25% of the Mount Pinatubo per year) applied to a climate warming following the RCP4.5 scenario from 7 earth system models (ESMs). The carbon and methane releases from permafrost were obtained from the PInc-PanTher model that was fed with the soil temperatures from the RCP4.5 or the GeoMIP G4 scenario, respectively. The estimates of the saved CO₂ and CH₄ losses and thus economical benefits from permafrost through stratospheric geoengineering in this paper are an important step for understanding and estimating the benefits and side effects of geoengineering. The method is sounds and novel. In addition, the topic is of interest to the general audience. Therefore, I recommend publication of this paper in Nature communications after my questions/concerns have been addressed.

Questions/concerns:

Is it sufficient to force the PInc-PanTher model with soil temperature? Wouldn't shortwave radiation also be important for soil processes?

lines 59/60: please explain why different ESMS show smaller or larger differences between the scenarios

line 129: what is meant by: this is slightly lower than the mitigating effects of G4 on permafrost soil C losses?

line 131: how do you know that the reduction in microbial decomposition is the main driver for stabilizing permafrost C stock?

lines 144/145: please explain why the models give different results.

line 148: isn't this result obvious if you force the PInc-PanTher model only with soil temperature? Or does this statement refer to the ESMS?

line 162: is deepest soil depth synonym with best soil model?

Minor comments and typos:

line 16: amount instead of amounts

line 27 and subsequent uses of ESM: ESM is normally used to as an abbreviation for Earth System Model, not Earth System Models. Thus, I suggest to write ESMS instead of ESM on line 41 and thereafter.

line 69: is close instead of are close

line 155: sentence seems incomplete

line 238: that only soil instead of that only on soil

line 239: change affects to affect

line 252: soil temperature is (add is)

Ulrike Lohmann

Reviewer #3 (Remarks to the Author):

Mitigation of Arctic permafrost carbon loss through stratospheric aerosol (SAI) geoengineering by Chen et al. uses data from GEO-MIP to in a permafrost carbon model to calculate avoided permafrost carbon feedback under stratospheric aerosol injection scheme. They use the avoided carbon emissions from permafrost in an integrated assessment model to calculate the economic impact of avoided

carbon release.

The effects of SAI on permafrost have been studied before, but the economic analysis is novel to my knowledge. Use of several models to predict the warming pattern of SAI makes it possible to assess the related uncertainty better than in previous studies. The manuscript is mostly clearly written. I think the economic analysis is interesting and a welcome contribution to literature as an estimate. However, there are potentially either some issues with calculating the impacts of CH₄ emissions or clarification needed in describing methods.

Major comments:

1) My main concern with the methodology of the manuscript is with making CH₄ emissions comparable with CO₂ emissions. The method description is somewhat unclear on this, so I don't know whether PAGE-ICE was using annual time-series of avoided CO₂ and CH₄ emissions or the aggregated CO₂ equivalent mean emission rate discussed in the main text. The following comment assumes that they are using a standard GWPs to convert CH₄ to CO₂e. This approach has several problems (e.g. Pierrehumbert, 2014, <https://doi.org/10.1146/annurev-earth-060313-054843>) that are not discussed or addressed in the manuscript. In my opinion, a better approach would be to use GWP* or some related concept that considers change in emission rate rather than pulse emissions (<https://doi.org/10.1016/j.ijggc.2012.11.028>, <https://doi.org/10.1038/nclimate2998>, doi: 10.1038/s41612-018-0026-8). Here the difference in CH₄ emissions between G4 and RCP4.5 seems to be rather constant.

2) The main policy-relevant point of the manuscript are the economic impacts of carbon released from the permafrost. To assess the importance of the findings, it would be good to know what are the economic impacts of avoided warming in G4 compared to RCP4.5. The PCF contributes further to this difference, but how much relatively? Also, some estimate on the relative contribution of avoided PCF to total avoided warming by SAI would be good.

Specific comments

Lines 22-23: Can you double-check this claim here. I didn't find the number 0.76 in the referenced article and six times larger than global average sounds surprisingly large.

Lines 36-67: Please, provide some references on moral hazard (decrease in motivation to decarbonise due to geoengineering).

Lines 42, 181 and 185: I think most countries have now NDCs instead of INDCs.

Line 42. I agree that RCP4.5 can arguably be labelled as most "policy-relevant", but I would not say the same of G4. More realistic scenarios have slower ramp up of sulphur, in my opinion.

Line 64: Explain please, what is the difference between PBP and NPP?

Lines 73-74: Can you make clear what is this range? Lowest and highest model results?

Line 84: Usually, Scandinavia excludes Greenland and Finland. I guess that this is not intention here?

Line 90: Does this "pure greenhouse gas forcing" include other forcings of RCP4.5. This is a bit confusing.

Line 150: "Well-replicated" compared to what? Do you mean that models predict similar patterns?

Lines 136-156: The uncertainty discussion could be improved. First, the current text suggests more "Sources of uncertainty in C estimates" than "Uncertainty in C estimates". It could be useful to summarize the uncertainty range here and state what you exactly are discussing here, and then try to show where it stems from.

If I understood correctly, the ESMs with deepest soils had least bias (Lines 260-261). Could you discuss, how the results would look, if these models were given more weight?

Overall, this section feels like a list of model differences, but it's hard to understand the relevance of these differences on the main results of this study.

I'm confused by the paragraph on ozone and NO_x. The text says first that the interactive chemistry in MIROC-ESM-CHEM creates differences, but then MIROC-ESM and MIROC-ESM-CHEM are similar to each other and different from BNU-ESM. So the interactive chemistry doesn't make much of a difference? If so, could you be more explicit here for better readability?

Does annual variability have any relevance here?

Line 155: What do you exactly mean by "diffusive effects"? Also "might be expected" sounds vague. Could you clarify the sentence?

Line 166: It seems that you are using methane's lifetime instead of its GPW value.

168: I think you mean CO₂-equivalent instead of CO₂-only (which could be understood to have only CO₂ emissions)

Line 173: I would suggest using the term carbon budget instead of "permissible anthropogenic C emission budget " and provide more references for it. I would also note that carbon release from permafrost is larger in RCP4.5 than in a scenario where global warming is limited under 2 degrees. Thus, your statement is somewhat exaggerating the amount of carbon released relative to 2-degree carbon budget.

Lines 174-175: The sentence about temperatures feels a bit out of place here.

Line 187: Could you combine the uncertainty from ESMs and from parametric uncertainty in PAGE-ICE? Now you have done them separately, but using the mean ESM temperature driver, Fig. 5 underestimates the total uncertainty.

Lines 214-215: The impact of prevention of permafrost carbon release is only a part of the full impact of SAI, and this sentence confuses that and the scope of this manuscript a bit. This would a good place to discuss how relevant PCF is in the big picture.

Line 221: developed instead of proposed?

Lines 269-271: Please, be more explicit what is exactly the input to PAGE-ICE. Does it take annual time-series of CO₂ and CO₂ from permafrost or some aggregated quantity?

Fig S6. Are the ranges given from ESMs? I'm a bit sceptical, how you derive 5% to 95% range with only seven models.

Table S1. These are differences between G4 and RCP4.5, right? Can you specify that?

Technical corrections:

Several places: ESM is used incorrectly for plural form instead of ESMs.

Fig. S6. Somewhat relevant elsewhere too, but especially here the axes lines are much more clearly visible than the actual plot lines.

Response to reviewer comments on “Mitigation of Arctic permafrost carbon loss through stratospheric aerosol geoengineering.” by Chen et al.

We are grateful to the reviewers for their constructive comments and valuable suggestions on this work, which greatly helped us improve the quality of the manuscript.

To address the significant points raised by the reviewers, we undertook substantial revisions of several key components of the manuscript, focusing on making statistical analysis of soil temperature between G4 and RCP4.5, refining the discussion of model differences and uncertainties, and clarifying the key inputs of our methods.

Below we address specific revisions in response to each reviewer comment. Line number references refer to those in the revised manuscript.

I. Response to Comments of Reviewer #1

This study addresses whether injection of sulfate aerosols into the stratosphere can mitigate permafrost carbon climate feedback. The authors use the Plnc-PanTher model of permafrost carbon–climate feedbacks driven by seven earth system models running the Geoengineering Model Intercomparison Project (GeoMIP) G4 stratospheric aerosol injection scheme to reduce radiative forcing under the Representative Concentration Pathway (RCP) 4.5 scenario. They find that permafrost carbon released as CO₂ is halved and as CH₄ by 40% under G4 compared with RCP4.5. They further assess that by mitigating ~13 Pg C in permafrost soils leads to about US\$8.0 trillion in economic value compared with RCP4.5.

The idea behind this study is very interesting and novel, where the authors convert potential carbon ‘not lost’ as a result of climate mitigation into economic value. This can be very useful in the policy point of view when discussing the application of geoengineering. And since SAI application is actually being discussed to be deployed as a pilot study, this kind of analysis is quite timely.

The general consensus of land surface models used in the CMIP5 experiments is that these models do not represent permafrost carbon very well... Due to this caveat, the authors chose to use Plnc-PanTher model driven by GeoMIP G4 as well as RCP4.5 scenario to quantify differences in CO₂ and CH₄ release. Although the general idea behind this study is interesting, I have concerns about the methods used in this study and the validity of using Plnc-PanTher model for this analysis.

We thank the Reviewer for sparing time to go through the manuscript, highlighting very important issues and providing valuable suggestions. We gave point-by-point response for each question and comment as follows:

First, permafrost carbon release greatly depends on temperature, particularly soil temperature. Therefore, the results in permafrost C losses depend solely on how different models exhibit soil temperatures under different scenarios. However, the authors only focus on permafrost soil C. There is a wider variety of range in soil temperatures and specifically different soil temperatures at depth. The results should focus more on why this is different. I don't see why getting the Ensemble mean would be important and interesting in this case without this discussion. Permafrost soil temperatures at different depths highly depend on how the models represent permafrost. But the authors fail to address and discuss this important point.

We now show explicitly the relationships between soil carbon loss and both soil and air temperatures in Fig. S6.

Fig. S6. Scatterplots of the yearly (2020 to 2069) differences in soil C loss (G4-RCP4.5) as a function of surface temperature differences (top row) and TSL differences (bottom row). The R2 and p values of the linear regressions are shown on each panel.

There are marked differences between models, but it is clear that TSL is a far better predictor of C losses than surface temperature forcing, with TSL typically accounting for 30% of variance in C, whereas surface T reflects only 10% of the variability, and is not a significant predictor for 5 of the 7 ESMs (Fig. S6). Thus although temperature is important it is far from sufficient in isolation to provide soil C balance. Hence we must disagree somewhat with the referee *“Therefore, the results in permafrost C losses depend solely on how different models exhibit soil temperatures under different scenarios.”*

In the totally revised section on Uncertainties we address the causes of ESM differences in soil C.

Line 172-180: *“Even though MIROC-ESM and MIROC-ESM-CHEM share the same land surface module (Table 1), TSL accounts for about 20% of variability in C loss for MIROC-ESM, but twice as much with MIROC-ESM-CHEM. This spans almost the full range of sensitivity of C losses to changes in TSL. The remarkable differences in TSL change (G4-*

RCP4.5) between the two MIROC models may also be seen by comparing the monthly TSL difference (RCP4.5-G4) for all the ESMs (Fig. S7). This suggests that natural variability over the 50-year simulations is as large as differences due to model formulation, and hence supports the Block et al., (2019)⁴⁸ view that initial conditions for Arctic sea ice and phases of the long period climate cycles driving temperature play very important roles in explaining across ESM variability, and strengthens the utility of the multi-model ensemble mean as a way of neutralizing these.”

Fig. S7. Monthly differences (G4-RCP4.5; units:°C) in TSL for each ESM over the 2020-2069 period.

The referee asks “why the ensemble mean is so important”. The answer is because the initial conditions are so important. The ESMs are not tuned for their long-period ocean-atmosphere-sea ice cycles to match the observations on decadal scales (it is important to note that these cycles are internally generated and may not be similar to observed climate, but can be used to tune models). Taking an ensemble negates the phase lags between the models and gives a more reasonable projection.

The final point of soil depth temperature profiles is addressed in the new Uncertainties section – we note it depends on many factors in addition to surface temperature forcing:

Line 181-198: “In the case of permafrost thermal state, the lowest soil boundary is a critical uncertainty affecting the simulation of permafrost⁴⁹. Earlier generations of land surface model typically had a lower boundary at 3m soil depth, but in general the deeper the soil layers the better the model is at simulating the thermal condition. Modeled snow depth and surface and soil temperature offsets vary widely amongst models⁵⁰. Some have substantial biases in their soil temperature simulations (Fig. S8), that are attributed mainly to inappropriate description of the surface (vegetation,

snow cover) and soil properties (soil texture, hydrology)⁵⁰. Snow cover plays an important role in modulating the variations of soil thermodynamics, and hence near-surface permafrost extent⁵¹. Several other factors such as differences in the treatment of soil organic matter, soil hydrology, surface energy calculations, model soil depth and vegetation also provide important controls on the simulated permafrost distribution⁵². Wang et al., (2016)⁵³ reviewed the performance of various land surface models, finding that models with better performance apply multilayer snow schemes. This allows them to simulate more realistic (stronger) insulation because they consider the snowpack’s vertical structure and variability. They calculate the energy and mass balance in each snow layer, are able to capture nonlinear profiles of snow temperature and can also account for thermal insulation within the snowpack such as when the upper layer thermally insulates the lower layers⁵⁴. These models also incorporate storage and refreezing of liquid water within the snow and parameterise wet-snow metamorphism, snow compaction and snow thermal conductivity (Table 1), which have been found to be among the most important processes for good snow depth and surface soil temperature simulation⁵³. These factors all interact and obscure relationships, and we find that soil C losses did not exhibit statistically significant relationships with snow cover in the ESMs.”

Table 1. Characteristics of 7 ESMs, the TSL and NPP outputs of which are used to drive permafrost C dynamics.

ESMs (Ref)	Atmospheric models	Land models	Resolution (lon×lat) ^b	Maximum soil depth (layers)	Dynamic vegetation	Nitrogen cycle
BNU-ESM ²⁰	CAM3.5	CoLM	128 × 64	3.6 m (10)	Yes	No
CanESM2 ²¹	AGCM4	CLASS2.7	128 × 64	4.1 m (3)	Yes	No
HadGEM2-ES ²²	AOGCM	MOSES2.2	192 × 145	3.0 m (4)	Yes	No
GISS-E2-R ²³	GISS-ModelE	Model II-LS	144 × 90	3.3 m (6)	No	No
MIROC-ESM ²⁴	MIROC-AGCM	MATSIRO	128 × 64	14.0 m (6)	Yes	No
^a MIROC-ESM-CHEM ²⁵	MIROCAGCM&CHASER	MATSIRO	128 × 64	14.0 m (6)	Yes	No
NorESM1-M ²⁶	CAM4-Oslo	CLM4.0	144 × 96	42.1 m (15)	No	Yes

(Continued)

ESMs (Ref)	Multiple snow layers	Snow insulation effect	Latent heat of soil water	Differing frozen/unfrozen soil thermal conductivity	Snow thermal conductivity (W m ⁻¹ K ⁻¹)	Snow density (ρ) (kg m ⁻³)
BNU-ESM ²⁰	Yes	Yes	Yes	Yes	f(ρ ₂)	f(snow depth; compaction)
CanESM2 ²¹	No	Yes	Yes	Yes	f(ρ ₂)	exp(-time/t)
HadGEM2-ES ²²	No	No	Yes	Yes	Fixed at 0.265	Fixed at 250
GISS-E2-R ²³	Yes	Yes	Yes	Yes	Fixed at 0.3	Fixed at 330
MIROC-ESM ²⁴	Yes	Yes	Yes	No	Fixed at 0.3	Fixed at 300
^a MIROC-ESM-CHEM ²⁵	Yes	Yes	Yes	No	Fixed at 0.3	Fixed at 300
NorESM1-M ²⁶	Yes	Yes	Yes	Yes	f(ρ ₂)	f(snow depth; compaction)

a. MIROC-ESM-CHEM shares the common features as MIROC-ESM, but with an additional atmospheric-chemistry component (CHASER).

b. all outputs are downscaled to 0.25° × 0.25° resolution and bias-corrected based on observations and reanalysis data.

Second, estimating permafrost C loss in CH₄ is an important aspect of this study. The tricky part of accurately projecting CH₄ emissions under permafrost thaw relies heavily on the projections of inundated area created under permafrost thaw. In PInc-PanTher, CH₄ emissions are a constant fraction of heterotrophic respiration from anoxic soils. Therefore, the CH₄ release from permafrost C estimated in this study seems a very crude estimation.

The referee is correct that accurately projecting CH₄ emissions relies heavily on the projections of inundated area created under permafrost thaw. Climate under greenhouse gas emissions is expected to be wetter than present, and climate under SAI will be drier, but these impacts are generally minor in comparison with temperature impacts – at least in analyses of e.g. glacier mass balance (Moore et al., 2019). This might be different for permafrost, but we are interested in understanding the potential response of just the forced, large-scale warming of soils and consequent increase in respiration (which may lead to increased CH₄ emissions). PInc-PanTher projects the inundated area based on soil C maps, assuming that all permafrost-affected peat soils are fully saturated and remain so under warming. Differing from the method of multiplying CH₄ flux density and inundated area (Melton et al., 2013), our calculation uses a constant scaling factor to relate inundated area and large-scale anoxic respiration rates to CH₄ fluxes. This simplified scaling approach is based on expert assessments (Schuur et al., 2013; 2015) and is still widely used in recent research (e.g. Schadel et al., 2016; Kessler, 2017; Gasser et al., 2018; Macias-Faurier et al., 2020). Previous estimates of the economic impact of permafrost carbon feedback, which are based on models with permafrost processes such as SiBCASA and JULES (Hope and Schaefer, 2016; Yumashev et al., 2019), also applied a scaling factor to estimate CH₄ emissions.

The drier SAI climates might mean that CH₄ losses could be relatively less under G4 than under RCP4.5, but no one has good enough models to demonstrate that at present. So the relative differences between G4 and RCP4.5 might be larger than we suggest. We rewrite this section, Line 132-148:

“Methane Losses

Aside from changes in soil C stocks, the other key concern from thawing permafrost is the magnitude of methane (CH₄) emissions as they have much larger warming potential per molecule than does CO₂³⁷. CH₄ emission rates depend on many factors, including in degree of waterlogging, in-situ carbon stocks, local warming, soil carbon-nitrogen ratios and biome³⁸. There are very few published estimates of the share of methane emissions in the present. It is certainly the case that particular regions will experience increases and other places decreases in wetlands and relative CH₄

emissions in the future. But it is also likely, since we are dealing with global averages, that these gains and losses will tend to cancel each other. PInc-PanTher projects the inundated area based on soil C maps, assuming that all Histel (permafrost-affected peat, which covers 1.4 million km²)³⁹ soils are fully saturated and remain so under warming. We calculate CH₄ emissions as a constant scaling factor to relate inundated area and large-scale anoxic respiration rates to CH₄ fluxes. This simplified scaling approach is based on expert assessments^{2,40} and has been widely used in recent research⁴¹⁻⁴⁴. Methane emissions are presumed to be 1.5%-3.5% of overall soil respiration rates^{40,45}, and we take a value of 2.3% here. PInc-PanTher simulations of the anoxic respiration rates over the period 2006-2010 are 1.2-1.6 Pg C yr⁻¹, and so the estimated range of CH₄ emissions is 28-37 Tg yr⁻¹, which is very close to the 15-40 Tg CH₄ yr⁻¹ estimates of current permafrost wetland CH₄ emissions⁴⁶. Since climate under G4 would be drier than under RCP4.5^{36,47}, assuming the same proportionate release of CH₄ under both scenarios might tend to minimize (RCP4.5-G4) differences, and so be regarded as a conservative estimate.”

References:

- Gasser, T. et al. Path-dependent reductions in CO₂ emission budgets caused by permafrost carbon release. *Nat. Geosci.* (2018) doi:10.1038/s41561-018-0227-0.
- Hope, C. & Schaefer, K. Economic impacts of carbon dioxide and methane released from thawing permafrost. *Nat. Clim. Chang.* (2016). doi:10.1038/nclimate2807
- Hugelius, G. et al. Estimated stocks of circumpolar permafrost carbon with quantified uncertainty ranges and identified data gaps. *Biogeosciences* (2014) doi:10.5194/bg-11-6573-2014.
- Kessler, L. Estimating the economic impact of the permafrost carbon feedback. *Clim. Chang. Econ.* (2017) doi:10.1142/S2010007817500087.
- Macias-Faurier, M. et al. Pleistocene arctic megafaunal ecological engineering as a natural climate solution?. *Philos. Trans. R. Soc. B Biol. Sci.* (2020) doi:10.1098/rstb.2019.0122.
- Melton, J. R. et al. Present state of global wetland extent and wetland methane modelling: Conclusions from a model inter-comparison project (WETCHIMP). *Biogeosciences* (2013) doi:10.5194/bg-10-753-2013.
- Moore, J. C., Yue, C., Zhao, L., Guo, X., Watanabe, S., & Ji, D. Greenland Ice Sheet Response to Stratospheric Aerosol Injection Geoengineering. *Earth's Future* (2019). doi:10.1029/2019EF001393
- Schadel, C. et al. Potential carbon emissions dominated by carbon dioxide from thawed permafrost soils. *Nat. Clim. Chang.* (2016) doi:10.1038/nclimate3054.
- Schuur, E. A. G. et al. Expert assessment of vulnerability of permafrost carbon to climate change. *Clim. Change* (2013). doi:10.1007/s10584-013-0730-7
- Schuur, E. A. G. et al. Climate change and the permafrost carbon feedback. *Nature* (2015) doi:10.1038/nature14338.
- Treat, C. C. et al. A pan-Arctic synthesis of CH₄ and CO₂ production from anoxic soil incubations. *Glob. Chang. Biol.* (2015) doi:10.1111/gcb.12875.
- Yumashev, D. et al. Climate policy implications of nonlinear decline of Arctic land permafrost and other cryosphere elements. *Nat. Commun.* (2019). doi:10.1038/s41467-019-09863-x

Third, as PInc-PanTher is a simple model that simulates soil carbon, the authors use NPP and NBP from CMIP5 models. Although, I wonder whether using NBP is a good choice in this case. NBP, by definition includes heterotrophic respiration in the term itself. If the authors are concerned about validity of using CMIP5 models in their soil C (and they are), they should not use this term. In addition, NBP includes ecosystem disturbance term such as fire, which in most of these models do not have interactive fire model, NBP is not a good choice in parameter comparison.

In fact, we only use TSL and NPP to drive the modified PInc-PanTher, and NBP is not an input parameter. The previous statement may be confusing, so we modified the description in the new manuscript:

Line 44-46: “To address this issue, we use a permafrost-based synthesis, the PInc-PanTher model modified to include CO₂ fertilization effects, to simulate the forced, large-scale permafrost C response to warming based on the bias-corrected soil temperatures (TSL) and net primary productivity (NPP) simulations of 7 ESMs (Table 1) forced with two common climate projections: the RCP4.5 and G4.”

Line 56: “Table 1. Characteristics of 7 ESMs, the TSL and NPP outputs of which are used to drive permafrost C dynamics.”

Line 84-90: “Most ESMs have lower NPP under G4 than RCP4.5 for the permafrost region because the cooling effect suppresses plant photosynthesis, the other models may have higher sensitivity to CO₂ fertilization than temperature, or the effects of diffuse light counteract the cooling effects (Fig. S3). But the net biological productivity (NBP; defined as NPP minus heterotrophic respiration and ecosystem disturbance) simulations of ESMs show small but positive differences for G4-RCP4.5 (Table S2), suggesting that, in contrast with the tropics, the slowing down of soil heterotrophic respiration due to cooling from SAI is the dominant response in the circum-Arctic permafrost region.”

Below are some of specific questions and comments I have throughout the manuscript.

- L12: *It is unclear what the hypothesis the authors are testing. Either clarify or revise this sentence.*

We revised this sentence in the manuscript as follows:

Line 10-15: “Increasing Earth’s albedo by the injection of sulfate aerosols into the stratosphere has been proposed as a way of offsetting some of the adverse effects of climate change. We examine this hypothesis in respect of permafrost carbon–climate feedbacks using the PInc-PanTher process model driven by seven earth system models

running the Geoengineering Model Intercomparison Project (GeoMIP) G4 stratospheric aerosol injection scheme to reduce radiative forcing under the Representative Concentration Pathway (RCP) 4.5 scenario.”

- L41-42: *To me, the motivation to make the comparison between G4 and RCP4.5 scenarios is unclear. The readers will benefit from some explanation.*

We modified this:

Line 46-50: “RCP4.5 is arguably the most policy-relevant scenario as the Nationally Determined Contributions (NDCs) greenhouse gas emissions framework would produce similar temperatures trajectories (Kitous and Keramidas, 2015), and G4 is one of the most “realistic” GeoMIP experiments which uses RCP4.5 as a control run and models stratospheric aerosol injection rather than solar insolation reduction. Furthermore, the amount of aerosol specified could plausibly be delivered (McClellan et al., 2012), and be relatively effective (Niemeier and Timmreck, 2015).”

References:

Kitous, A. & Keramidas, K. Analysis of scenarios integrating the INDCs. Jt. Res. Centre, Sevilla, Spain (2015).

McClellan, J., Keith, D. W. & Apt, J. Cost analysis of stratospheric albedo modification delivery systems. Environ. Res. Lett. (2012) doi:10.1088/1748-9326/7/3/034019.

Niemeier, U. & Timmreck, C. What is the limit of climate engineering by stratospheric injection of SO₂? Atmos. Chem. Phys. (2015) doi:10.5194/acp-15-9129-2015.

- L46: *The authors should give reasoning why they are using Plnc-PanTher model rather than permafrost soil C release directly from ESMs.*

In the introduction Line 41-46, we now say: “However, permafrost C cycle dynamics are poorly represented in CMIP5 models as they were not developed for permafrost soils and do not report vertically resolved soil layers¹⁹.

To address this issue, we simulate the large-scale permafrost C response to warming using the Plnc-PanTher model modified to include CO₂ fertilization effects, forced by the bias-corrected soil temperatures (TSL) and net primary productivity (NPP) simulations of 7 ESMs (Table 1) under two climate projections: RCP4.5 and G4.”

- L51: *Is this circum-Arctic permafrost?*

Yes, we revise several places for more accurate expression. For example,

Line 62-63: “Fig. 1 shows the circum-Arctic permafrost C loss simulated by PInc-PanTher when driven by the bias-corrected TSL and NPP of seven ESMs (Methods).”

Line 96-97: “Fig. 1. (a) cumulative losses of circum-Arctic permafrost C storage from PInc-PanTher between Apr 2006 and 2100; each curve is the mean of two methods of calculation (Fig. S1; Methods).”

- Section ‘Permafrost C loss’: What about the difference in soil T between G4 and RCP4.5? What is the magnitude? Also, why didn't the authors compare the difference between RCP8.5+Geoengineering with RCP4.5 where the temperature levels are similar? The authors should show temperature-time series plot and discuss why there is such large variability in permafrost C loss across the models.

- What about the difference in soil T between G4 and RCP4.5?

We make a new figure to show this difference

Fig. S7. Monthly differences (G4-RCP4.5; units:°C) in TSL for each ESM over the 2020-2069 period.

- What is the magnitude?

Line 74-78: “Between 2020 and 2069, warming of the upper 3 m permafrost soils are 1.3 ± 1.2 and 2.2 ± 1.2 °C under G4 and RCP4.5, respectively, and the ensemble mean PInc-PanTher simulations are net soil C losses throughout the experimental period

under both scenarios. Anomalies (G4-RCP4.5) exhibit much less scatter in permafrost C preserved (-13.3 ± 4.1 Pg) due to the well-replicated cooling effect of SAI (-0.9 ± 0.4 °C) among all ESMs that have run G4³²(Fig. S2), indicating that scenario forcing dominates model differences.”

Fig. S2. Maps of soil T change between 2020 and 2069 under G4 and RCP4.5 scenarios, averaged from upper 3 meters.

- why didn't the authors compare the difference between RCP8.5+Geoengineering with RCP4.5 where the temperature levels are similar?

RCP8.5 is precluded from policy given the Paris NDC pledges, so as we are interested in policy relevant scenarios we don't focus on RCP8.5. In 2020 these pledges are supposed to be strengthened. There are enormous and internationally agreed benefits of not doing RCP8.5 emissions (including ocean acidification that SAI cannot mitigate). We do recognize that permafrost damage under RCP8.5 will be more severe, but we do not think that is a priority given that this is universally, or at least widely, agreed already. No one is proposing doing SAI to avoid mitigation.

- The authors should show temperature-time series plot and discuss why there is such large variability in permafrost C loss across the models

We now show temperature-time series plot in the new Fig. 1 and discuss the ESM differences in the new Uncertainties section:

Line 159-216:

“Sources of Uncertainty in C estimates

The ESMs differ in very many ways and Table 1 lists only a few of them. Moore et al.,

(2020)³⁶ found that 4 of the same models as used here also produce a similar spread of differences (RCP4.5-G4) in mass balance for Greenland as found for C release from permafrost. This results from combinations of reasons: differences in the sensitivities of ESMs to greenhouse gas and SAI forcing means that simulated changes in AMOC, seasonal sea ice, cloud cover, specific humidity and hence longwave radiative absorption and accumulation rates all differ. A recent analysis⁴⁸ of ESM differences in the Arctic across 13 CMIP5 models including BNU-ESM, MIROC, NorESM1-M, CanESM2, GISS-E2-R, explains it largely in terms of differing estimates of surface albedo and Planck feedbacks which show the largest intermodel differences. Their analysis indicates that these differences arise not only from different degrees of simulated Arctic warming but also are partly related to the large differences in initial sea ice cover and surface temperatures⁴⁸. Fig. S6 shows the differences (G4-RCP4.5) between soil C annual losses as a function of differences in TSL and surface temperatures. There are marked differences between models, but it is clear that TSL is a far better predictor of C losses than surface temperature forcing, with TSL typically accounting for 30% of variance in C, whereas surface T reflects only 10% of the variability, and is not a significant predictor for 5 of the 7 ESMs (Fig. S6). Even though MIROC-ESM and MIROC-ESM-CHEM share the same land surface module (Table 1), TSL accounts for about 20% of variability in C loss for MIROC-ESM, but twice as much with MIROC-ESM-CHEM. This spans almost the full range of sensitivity of C losses to changes in TSL. The remarkable differences in TSL change (G4-RCP4.5) between the two MIROC models may also be seen by comparing the monthly TSL difference (RCP4.5-G4) for all the ESMs (Fig. S7). This suggests that natural variability over the 50-year simulations is as large as differences due to model formulation, and hence supports the Block et al., (2019)⁴⁸ view that initial conditions for Arctic sea ice and phases of the long period climate cycles driving temperature play very important roles in explaining across ESM variability, and strengthens the utility of the multi-model ensemble mean as a way of neutralizing these.

In the case of permafrost thermal state, the lowest soil boundary is a critical uncertainty affecting the simulation of permafrost⁴⁹. Earlier generations of land surface model typically had a lower boundary at 3m soil depth, but in general the deeper the soil layers the better the model is at simulating the thermal condition. Modeled snow depth and surface and soil temperature offsets vary widely amongst models⁵⁰. Some have substantial biases in their soil temperature simulations (Fig. S8), that are attributed mainly to inappropriate description of the surface (vegetation, snow cover) and soil properties (soil texture, hydrology)⁵⁰. Snow cover plays an important role in modulating the variations of soil thermodynamics, and hence near-surface permafrost extent⁵¹. Several other factors such as differences in the treatment of soil organic matter, soil hydrology, surface energy calculations, model soil depth and vegetation also provide important controls on the simulated permafrost distribution⁵². Wang et al., (2016)⁵³ reviewed the performance of various land surface models, finding that models with better performance apply multilayer snow schemes. This allows them to simulate more realistic (stronger) insulation because they consider the snowpack's vertical structure and variability. They calculate the energy and mass

balance in each snow layer, are able to capture nonlinear profiles of snow temperature and can also account for thermal insulation within the snowpack such as when the upper layer thermally insulates the lower layers⁵⁴. These models also incorporate storage and refreezing of liquid water within the snow and parameterise wet-snow metamorphism, snow compaction and snow thermal conductivity (Table 1), which have been found to be among the most important processes for good snow depth and surface soil temperature simulation⁵³. These factors all interact and obscure relationships, and we find that soil C losses did not exhibit statistically significant relationships with snow cover in the ESMs.

In comparison with the land surface models, differences in the treatment of stratospheric aerosol physics and chemistry⁴⁷ probably plays a minor role. For example, HadGEM2-ES adopts a stratospheric aerosol schemes to simulate the sulfate aerosol optical depth (AOD). BNU-ESM and MIROC-ESM use the prescribed meridional distribution of AOD recommended by the GeoMIP protocol⁵⁵. MIROC-ESM and MIROC-ESM-CHEM share most processes except for an interactive atmospheric chemistry module in MIROC-ESM-CHEM⁵⁶. But comparing the BNU-ESM results in Fig.1 with MIROC-ESM and MIROC-ESM-CHEM (which share similar SAI schemes), we see that BNU-ESM produces much lower soil C losses than the other two which have very similar results. This is likely because BNU and both MIROC models have different land surface models, soil depths and different ways of treating the snow cover (Table 1). For the purposes of permafrost stability the general pattern of cooling produced seems dominant as soil temperatures appear the single most critical variable in determining the differences in response between G4 and RCP4.5. The patterns of cooling are well-replicated by all ESMs that have run G4³² (Fig. S2).

The lifespans of stratospheric sulfur aerosols in different regions and altitudes vary from a few months to several years⁵⁷. Since the G4 experiment does not remove CO₂ from the atmosphere, if radiative forcing by SAI were to suddenly cease, there would be a compensatory rapid global temperature rise, the "termination shock"⁵⁸. The rise in surface air temperatures following termination causes a strong air-soil temperature gradient that drives heat into the soil generating a sudden increase in permafrost emissions from 2070-2089 which partially offsets the carbon losses avoided during the G4 implementation (Fig. S9, Table S1). Hopefully, such an unplanned termination strategy would not be employed in reality⁵⁹."

References:

48. Block, K., Schneider, F. A., Mülmenstädt, J., Salzmann, M., & Quaas, J. Climate models disagree on the sign of total radiative feedback in the Arctic. *Tellus, Ser. A Dyn. Meteorol. Oceanogr.* (2019). doi:10.1080/16000870.2019.1696139
49. Nicolsky, D. J., Romanovsky, V. E., Alexeev, V. A. & Lawrence, D. M. Improved modeling of permafrost dynamics in a GCM land-surface scheme. *Geophys. Res. Lett.* (2007) doi:10.1029/2007GL029525.
50. Wang, W. et al. Diagnostic and model dependent uncertainty of simulated Tibetan permafrost area. *Cryosphere* (2016) doi:10.5194/tc-10-287-2016.
51. Park, H., Fedorov, A. N., Zheleznyak, M. N., Konstantinov, P. Y. & Walsh, J. E. Effect of snow cover on pan-Arctic permafrost thermal regimes. *Clim. Dyn.* (2015) doi:10.1007/s00382-014-2356-5.
52. Marchenko, S. & Etzelmüller, B. Permafrost: Formation and Distribution, Thermal and Mechanical Properties. in *Treatise on Geomorphology* (2013). doi:10.1016/B978-0-12-374739-6.00207-4.
53. Wang, W. et al. Evaluation of air-soil temperature relationships simulated by land surface models during winter across the permafrost region. *Cryosphere* (2016). doi:10.5194/tc-10-1721-2016

54. Dutra, E., Viterbo, P., Miranda, P. M. A. & Balsamo, G. Complexity of snow schemes in a climate model and its impact on surface energy and hydrology. *J. Hydrometeorol.* (2012) doi:10.1175/JHM-D-11-072.1.
55. Kashimura, H. et al. Shortwave radiative forcing, rapid adjustment, and feedback to the surface by sulfate geoengineering: Analysis of the Geoengineering Model Intercomparison Project G4 scenario. *Atmos. Chem. Phys.* (2017). doi:10.5194/acp-17-3339-2017
56. Pitari, G. et al. Stratospheric ozone response to sulfate geoengineering: Results from the geoengineering model intercomparison project (GeoMip). *J. Geophys. Res.* (2014). doi:10.1002/2013JD020566
57. Rasch, P. J., Crutzen, P. J. & Coleman, D. B. Exploring the geoengineering of climate using stratospheric sulfate aerosols: The role of particle size. *Geophys. Res. Lett.* (2008). doi:10.1029/2007GL032179
58. Jones, A. et al. The impact of abrupt suspension of solar radiation management (termination effect) in experiment G2 of the Geoengineering Model Intercomparison Project (GeoMIP). *J. Geophys. Res. Atmos.* (2013). doi:10.1002/jgrd.50762
59. Parker, A. & Irvine, P. J. The Risk of Termination Shock From Solar Geoengineering. *Earth's Futur.* (2018). doi:10.1002/2017EF000735

- L53-55: This is interesting. How? When you look at the difference in global C cycling (or even soil C) across the CMIP5 models, there is a very large spread. Does this mean that soil T is not playing much effect on this? Can the authors go into this a bit in more detail? Again, showing the soil T of all the models would be helpful.

We show the soil T of all ESMs and the relationship between ΔT and ΔC in new Fig.1. The soil temperature in the previous manuscript was slightly inaccurate and is not an area-weighted average of the circum-Arctic permafrost region. After recalculation, there is a more obvious relationship between soil T change (ΔT) and soil C loss (ΔC).

Fig. 1. (a) cumulative losses of circum-Arctic permafrost C storage from PInc-PanTher between 2006 and 2100; each curve is the mean of two methods of calculation (Fig. S1; Methods). (b) mean annual temperature in the upper 3 m of soils, derived from bias-corrected ESM outputs (Table 1), solid curves are G4, dashed are RCP4.5. (c) scatter plot of permafrost C loss and soil T change during SAI implementation (2020-2069). Positive values indicate losses of soil C stocks to atmosphere.

Projected soil C losses are roughly linearly proportional to soil temperature changes across the two scenarios ($R^2=0.697$, $p<0.01$), and this apparent linearity is particularly evident for G4-RCP4.5 (with only one outlier HadGEM2-ES).

Line 67-71: “Projected C losses are roughly linearly proportional to changes in soil temperature, and each 1°C warming in the Arctic permafrost would result in approximately 13 Pg C loss; the y-intercept indicates that the Arctic permafrost, if maintained in current state, would remain a weak carbon sink. MIROC-ESM and MIROC-ESM-CHEM, with simulations of warming above 3°C, produce severe soil C losses, while GISS-E2-R with minor soil temperature change produces net soil C gains under both scenarios before 2070.”

Line 80-82: “Most simulations of difference between G4 and RCP4.5 scenarios fit well with the linear relationship between ΔC and ΔT (Fig. 1), and the CanESM2 and MIROC-ESM-CHEM simulations give the most and least significant difference, respectively.”

Line 90-94: “The outliers in Fig.1 may be explained by the time-varying soil C input flux related to NPP. For example, HadGEM2-ES which simulates moderate warming and the strongest cooling effects by SAI, produces net soil C gain, due to the sharp increase in NPP (Fig. S3). And the NorESM1-M with a nitrogen cycle but no dynamic vegetation (Table 1) simulates constant NPP throughout the period and gives relatively high soil C loss.”

However, in the year by year responses (Fig. S6 given earlier), differences in TSL only explain about 30% of variability in soil C losses. And the comparison of MIROC-ESM and MIROC-ESM-CHEM suggests that natural variability over the 50-year simulations is as large as differences due to model formulation.

- L61-62: I am very confused about this statement because only two out of the seven models show lower NPP under G4 compared to RCP4.5 (Fig.s2). I suspect that the other models have higher sensitivity to CO2 fertilization over temperature. This needs to be pointed out.

The previous Fig. S2 is not very clear, we have redrawn it and limited the time to 2069. For BNU-ESM, GISS-E2-R, HadGEM2-ES and MIROC-ESM, the NPP is lower under G4 compared to RCP4.5. We modified as suggested:

Line 84-87: “Most ESMs have lower NPP under G4 than RCP4.5 for the permafrost region because the cooling effect suppresses plant photosynthesis, the other models may have higher sensitivity to CO₂ fertilization than temperature, or the effects of diffuse light counteract the cooling effects (Fig. S3). ”

Fig. S3. Net primary productivity in permafrost region before (top) and after (bottom) bias correction for RCP4.5 (orange) and G4 (blue).

- L63-64: *I don't understand this transition. Please consider revising.*

We revised as follow:

Line 84-90: “Most ESMs have lower NPP under G4 than RCP4.5 for the permafrost region because the cooling effect suppresses plant photosynthesis, the other models may have higher sensitivity to CO₂ fertilization than temperature, or the effects of diffuse light counteract the cooling effects (Fig. S3). But the net biological productivity (NBP; defined as NPP minus heterotrophic respiration and ecosystem disturbance) simulations of ESMs show small but positive differences for G4-RCP4.5 (Table S2), suggesting that, in contrast with the tropics, the slowing down of soil heterotrophic respiration due to cooling from SAI is the dominant response in the circum-Arctic permafrost region.”

- L70-72: *This is understandable because first, vegetation response is quite constrained to temperature in high latitude ecosystems and the growing season is very short. In other regions, this may not be the case as vegetation response can be due to multiple factors such as precipitation, CO₂ fertilization, and radiation. But I do not understand the authors' reasoning why this is important. Is it due to the C input to soil? Please explain.*

Yes, agreed. We modified this paragraph (same as last answer).

- L107-108: *Q10 is temperature sensitivity of decomposition, not decomposition rate itself. In this case, the authors may be better off directly showing the rate of decomposition (k) or heterotrophic respiration than Q10. To me, the two lines look very similar and without any statistical analysis, it could be an overstatement of the authors to say that the two simulations exhibit different soil temperatures.*

Yes, Q_{10} is just the temperature function, and the k values depend on the horizon types and pools in Plnc-PanTher. Heterotrophic respiration may not be suitable for calculations on a monthly scale due to the nonlinear relationship between k and temperature.

In new Fig.3, we still use line graphs to show the monthly changes in soil temperature at each layer in the Arctic permafrost region. And we use histograms with error bars to show the rise in soil temperature between 2020 and 2069 under the RCP4.5 and G4.

Line 119-127: “The exponential relationship between decomposition rate and soil temperature indicates that soil respiration mainly occurs in summer, and therefore a cooling by one Celsius has a greater effect on permafrost carbon preservation if it occurs in a warmer month than a cold one. This implies using SAI to target permafrost preservation need only be done in the summer season. We calculate the average monthly soil temperature and also the changes between 2020 and 2069 at different soil depths (Fig. 3). The upper meter is where most of the permafrost is predicted to be thawing under warming, and decomposition occurs from June to October when there are also the obvious differences of temperature rise between G4 and RCP4.5 (Fig. 3). At 1-3 m depth, soil temperature rise between 2020 and 2069 is fairly uniform, as is the cooling effect of G4. The summer temperature warming wave is increasingly lagged at deeper soil depths and decomposition occurs as late as January.”

Fig. 3. Histogram with error bars of monthly soil temperature change at different soil depths (as labeled), and mean temperatures in the soil (shown as lines scaled by right

y-axis) for RCP4.5 (orange) and G4 (blue) over the period 2020-2069.

- L112-114: *Could the authors emphasize here that because SAI is a radiation management method it is expected that it is not so effective in the winter months where the length of solar period is shorter?*

One might suppose that, but it is a simplistic assumption that does not take into account transport pathways in the ocean-atmosphere system, nor the physical permafrost environment. Ji et al (2018) show that extremes vary with latitude and type of solar geoengineering, The impacts of snow season and melt dates also strongly affect permafrost and river runoff (Wei et al, 2018).

References:

Ji, D. et al. Extreme temperature and precipitation response to solar dimming and stratospheric aerosol geoengineering. Atmos. Chem. Phys. (2018). doi:10.5194/acp-18-10133-2018

Wei, L., Ji, D., Miao, C., Muri, H. & Moore, J. C. Global streamflow and flood response to stratospheric aerosol geoengineering. Atmos. Chem. Phys. (2018). doi:10.5194/acp-18-16033-2018

- L118: *Why do the authors assume decomposition does not occur below 0C? Is this because of how PInc-PanTher operates?*

Yes, PInc-PanTher uses a truncated Q_{10} function, which assumes zero respiration at or below the freezing point. PInc-PanTher is a simplified model approach for projecting soil C losses in response to soil warming throughout the soil column.

We modified the Method description of equation 1 in the manuscript:

Line 322-324: "PInc-PanTher uses a truncated Q_{10} function, which assumes zero soil respiration when soil temperature is at, or below, the freezing point⁸, which is only an approximation to reality, but removes spurious winter decomposition rates."

- Section 'Uncertainties in C estimates': *The authors fail to recognize or discuss the differences in how different models simulate physical processes associated to permafrost. This has a large impact on soil temperature, particularly deep soil temperature. As a result, the difference in permafrost recognition across models are really causing the large variability. This is a very important aspect, so should be discussed here.*

Yes, this deserves more treatment than we have given it. So we have re-written the section on Uncertainties. We have also added more details on each models' land surface and snow scheme in a revised Table 1. Furthermore we analyze each ESM C response to differences in both soil and surface temperature in a new supplementary

figures S6 and S7.

Line 159-216:

“Sources of Uncertainty in C estimates

The ESMs differ in very many ways and Table 1 lists only a few of them. Moore et al., (2020)³⁶ found that 4 of the same models as used here also produce a similar spread of differences (RCP4.5-G4) in mass balance for Greenland as found for C release from permafrost. This results from combinations of reasons: differences in the sensitivities of ESMs to greenhouse gas and SAI forcing means that simulated changes in AMOC, seasonal sea ice, cloud cover, specific humidity and hence longwave radiative absorption and accumulation rates all differ. A recent analysis⁴⁸ of ESM differences in the Arctic across 13 CMIP5 models including BNU-ESM, MIROC, NorESM1-M, CanESM2, GISS-E2-R, explains it largely in terms of differing estimates of surface albedo and Planck feedbacks which show the largest intermodel differences. Their analysis indicates that these differences arise not only from different degrees of simulated Arctic warming but also are partly related to the large differences in initial sea ice cover and surface temperatures⁴⁸. Fig. S6 shows the differences (G4-RCP4.5) between soil C annual losses as a function of differences in TSL and surface temperatures. There are marked differences between models, but it is clear that TSL is a far better predictor of C losses than surface temperature forcing, with TSL typically accounting for 30% of variance in C, whereas surface T reflects only 10% of the variability, and is not a significant predictor for 5 of the 7 ESMs (Fig. S6). Even though MIROC-ESM and MIROC-ESM-CHEM share the same land surface module (Table 1), TSL accounts for about 20% of variability in C loss for MIROC-ESM, but twice as much with MIROC-ESM-CHEM. This spans almost the full range of sensitivity of C losses to changes in TSL. The remarkable differences in TSL change (G4-RCP4.5) between the two MIROC models may also be seen by comparing the monthly TSL difference (RCP4.5-G4) for all the ESMs (Fig. S7). This suggests that natural variability over the 50-year simulations is as large as differences due to model formulation, and hence supports the Block et al., (2019)⁴⁸ view that initial conditions for Arctic sea ice and phases of the long period climate cycles driving temperature play very important roles in explaining across ESM variability, and strengthens the utility of the multi-model ensemble mean as a way of neutralizing these.

In the case of permafrost thermal state, the lowest soil boundary is a critical uncertainty affecting the simulation of permafrost⁴⁹. Earlier generations of land surface model typically had a lower boundary at 3m soil depth, but in general the deeper the soil layers the better the model is at simulating the thermal condition. Modeled snow depth and surface and soil temperature offsets vary widely amongst models⁵⁰. Some have substantial biases in their soil temperature simulations (Fig. S8), that are attributed mainly to inappropriate description of the surface (vegetation, snow cover) and soil properties (soil texture, hydrology)⁵⁰. Snow cover plays an important role in modulating the variations of soil thermodynamics, and hence near-surface permafrost extent⁵¹. Several other factors such as differences in the treatment of soil organic matter, soil hydrology, surface energy calculations, model soil depth and vegetation also provide important controls on the simulated permafrost distribution⁵².

Wang et al., (2016)⁵³ reviewed the performance of various land surface models, finding that models with better performance apply multilayer snow schemes. This allows them to simulate more realistic (stronger) insulation because they consider the snowpack's vertical structure and variability. They calculate the energy and mass balance in each snow layer, are able to capture nonlinear profiles of snow temperature and can also account for thermal insulation within the snowpack such as when the upper layer thermally insulates the lower layers⁵⁴. These models also incorporate storage and refreezing of liquid water within the snow and parameterise wet-snow metamorphism, snow compaction and snow thermal conductivity (Table 1), which have been found to be among the most important processes for good snow depth and surface soil temperature simulation⁵³. These factors all interact and obscure relationships, and we find that soil C losses did not exhibit statistically significant relationships with snow cover in the ESMs.

In comparison with the land surface models, differences in the treatment of stratospheric aerosol physics and chemistry⁴⁷ probably plays a minor role. For example, HadGEM2-ES adopts a stratospheric aerosol schemes to simulate the sulfate aerosol optical depth (AOD). BNU-ESM and MIROC-ESM use the prescribed meridional distribution of AOD recommended by the GeoMIP protocol⁵⁵. MIROC-ESM and MIROC-ESM-CHEM share most processes except for an interactive atmospheric chemistry module in MIROC-ESM-CHEM⁵⁶. But comparing the BNU-ESM results in Fig.1 with MIROC-ESM and MIROC-ESM-CHEM (which share similar SAI schemes), we see that BNU-ESM produces much lower soil C losses than the other two which have very similar results. This is likely because BNU and both MIROC models have different land surface models, soil depths and different ways of treating the snow cover (Table 1). For the purposes of permafrost stability the general pattern of cooling produced seems dominant as soil temperatures appear the single most critical variable in determining the differences in response between G4 and RCP4.5. The patterns of cooling are well-replicated by all ESMs that have run G4³² (Fig. S2).

The lifespans of stratospheric sulfur aerosols in different regions and altitudes vary from a few months to several years⁵⁷. Since the G4 experiment does not remove CO₂ from the atmosphere, if radiative forcing by SAI were to suddenly cease, there would be a compensatory rapid global temperature rise, the "termination shock"⁵⁸. The rise in surface air temperatures following termination causes a strong air-soil temperature gradient that drives heat into the soil generating a sudden increase in permafrost emissions from 2070-2089 which partially offsets the carbon losses avoided during the G4 implementation (Fig. S9, Table S1). Hopefully, such an unplanned termination strategy would not be employed in reality⁵⁹."

- Throughout: In this version of the manuscript, the way it is written can make the reader very confused about what the authors did. It often sounds like the authors are actually using the CMIP5 simulations, but actually what the authors did was to use soil T data extracted from the CMIP5 models to force PInc-PanTher model. This needs to be better portrayed in the writing.

Sorry for any confusion. We clarify the methodology.

E.g. in the abstract: “We examine this hypothesis in respect of permafrost carbon–climate feedbacks using the PInc-PanTher process model driven by seven earth system models running the Geoengineering Model Intercomparison Project (GeoMIP) G4 stratospheric aerosol injection scheme to reduce radiative forcing under the Representative Concentration Pathway (RCP) 4.5 scenario”

We have looked through the manuscript trying to find where confusion might arise on this point.

In the introduction Line 41-46: we now say: “However, permafrost C cycle dynamics are poorly represented in CMIP5 models as they were not developed for permafrost soils and do not report vertically resolved soil layers. To address this issue, we simulate the large-scale permafrost C response to warming using the PInc-PanTher model modified to include CO₂ fertilization effects, forced by the bias-corrected soil temperatures (TSL) and net primary productivity (NPP) simulations of 7 ESMs (Table 1) under two climate projections: RCP4.5 and G4.”

In the old “Methods” we say: “We use all ESMs that have monthly mean values of soil temperature and annual NPP, available for G4, RCP4.5 as well as the historical period (Table 1).”

We modify this Line 326-327: “We drive the **modified PInc-PanTher with** all ESMs that have the monthly mean values of soil temperature and annual NPP, available for G4, RCP4.5 as well as the historical period (Table 1).”

II. Response to Comments of Reviewer #2

The paper by Chen et al. addresses the impact of stratospheric geoengineering on mitigation of carbon and methane loss from Arctic permafrost. Stratospheric geoengineering was obtained from the GeoMIP G4 stratospheric injection scheme scenario (injection approximately 25% of the Mount Pinatubo per year) applied to a climate warming following the RCP4.5 scenario from 7 earth system models (ESMs). The carbon and methane releases from permafrost were obtained from the Plnc-PanTher model that was fed with the soil temperatures from the RCP4.5 or the GeoMIP G4 scenario, respectively. The estimates of the saved CO₂ and CH₄ losses and thus economical benefits from permafrost through stratospheric geoengineering in this paper are an important step for understanding and estimating the benefits and side effects of geoengineering. The method is sounds and novel. In addition, the topic is of interest to the general audience. Therefore, I recommend publication of this paper in Nature communications after my questions/concerns have been addressed.

We would like to thank the Reviewer for the appreciation of the main advances of our work. For each question and comment, we gave point-by-point response and made precise additions and careful revisions to this manuscript. The content is as follows:

Questions/concerns:

Is it sufficient to force the Plnc-PanTher model with soil temperature? Wouldn't shortwave radiation also be important for soil processes?

In the **standard** Plnc-PanTher model (Koven et al., 2015), projections of permafrost carbon release follow set decomposition trajectories, as a function of soil temperature.

The changes in shortwave radiation directly caused by SAI are rather small, changes of 1-2W/m² or about 1% of the solar insolation (Kashimura et al., 2017). The permafrost soil temperature is determined by the energy balance composed of radiation (short and long wave), turbulent heat fluxes (sensible and latent), and heat flux into the ground, snow, or water bodies (Boike et al., 2012). Therefore, we believe that a large part of the effects of shortwave radiation is reflected in soil temperature.

In addition, our **modified** Plnc-PanTher model is forced by temperature of soil layers (TSL) and net primary productivity (NPP), because McGuire et al. (2018) point out that ignoring response of vegetative productivity to climate change may cause overestimation of soil C release. Changes in soil C stocks depend on soil heterotrophic respiration and vegetation input flux, and soil respiration depends on TSL, while the

input flux is proportional to NPP. (Todd-Brown et al., 2013)

We took care to use a bias correction method that conserves trends to the ESM input data while ensuring that the initial state is consistent with observation and reanalysis data. We also examined various other ESM fields, such as snow, for impacts on soil C funding no significant relationship, and hence we simply use the model fields we find to be statistically useful.

References:

Boike, J. et al. Permafrost – physical aspects, carbon cycling, databases and uncertainties. in Recarbonization of the Biosphere: Ecosystems and the Global Carbon Cycle (2012). doi:10.1007/978-94-007-4159-1_8.

Kashimura, H. et al. Shortwave radiative forcing, rapid adjustment, and feedback to the surface by sulfate geoengineering: Analysis of the Geoengineering Model Intercomparison Project G4 scenario. Atmos. Chem. Phys. (2017) doi:10.5194/acp-17-3339-2017.

Koven, C. D. et al. A simplified, data-constrained approach to estimate the permafrost carbon-climate feedback. Philos. Trans. R. Soc. A Math. Phys. Eng. Sci. (2015). doi:10.1098/rsta.2014.0423

McGuire, A. D. et al. Dependence of the evolution of carbon dynamics in the northern permafrost region on the trajectory of climate change. Proc. Natl. Acad. Sci. (2018). doi:10.1073/pnas.1719903115

Todd-Brown, K. E. O. et al. Causes of variation in soil carbon simulations from CMIP5 Earth system models and comparison with observations. Biogeosciences (2013) doi:10.5194/bg-10-1717-2013.

lines 59/60: please explain why different ESMs show smaller or larger differences between the scenarios

We modified the “Permafrost C loss” section as follow:

Line 62-71: “Fig. 1 shows the circum-Arctic permafrost C loss simulated by PInc-PanTher when driven by the bias-corrected TSL and NPP of seven ESMs (Methods). There is considerable across-model spread, but consistent and significant differences at the 95% level between C losses under the two scenarios (Table S1). Between 2020 and 2069, PInc-Panther simulations of soil C change, driven by outputs of 7 ESMs for the RCP4.5 projection, varied from 19.4 Pg C gain to 51.4 Pg C loss (mean 24.6 Pg C loss), while under G4 the ensemble mean was 11.3 Pg C loss (range: 28.7 Pg C gain to 43.7 Pg C loss). Projected C losses are roughly linearly proportional to changes in soil temperature, and each 1°C warming in the Arctic permafrost would result in approximately 13 Pg C loss; the y-intercept indicates that the Arctic permafrost, if maintained in current state, would remain a weak carbon sink. MIROC-ESM and MIROC-ESM-CHEM, with simulations of warming above 3°C, produce severe soil C losses, while GISS-E2-R with minor soil temperature change produces net soil C gains under both scenarios before 2070.”

Line 74-82: “Between 2020 and 2069, warming of the upper 3 m permafrost soils are

1.3±1.2 and 2.2±1.2 °C under G4 and RCP4.5, respectively, and the ensemble mean Plnc-PanTher simulations are net soil C losses throughout the experimental period under both scenarios. Anomalies (G4-RCP4.5) exhibit much less scatter in permafrost C preserved (-13.3±4.1 Pg) due to the well-replicated cooling effect of SAI (-0.9±0.4 °C) among all ESMs that have run G4³² (Fig. S2), indicating that scenario forcing dominates model differences. The difference is significant at the 95% level for 5 of the 7 models (Table S1). Thus, the ensemble mean difference G4-RCP4.5 is a halving of soil C released in the circum-Arctic permafrost region with SAI geoengineering. Most simulations of difference between G4 and RCP4.5 scenarios fit well with the linear relationship between ΔC and ΔT (Fig. 1), and the CanESM2 and MIROC-ESM-CHEM simulations give the most and least significant difference, respectively.”

Fig. 1. (a) cumulative losses of circum-Arctic permafrost C storage from Plnc-PanTher between 2006 and 2100; each curve is the mean of two methods of calculation (Fig. S1; Methods). (b) mean annual temperature in the upper 3 m of soils, derived from bias-corrected ESM outputs (Table 1), solid curves are G4, dashed are RCP4.5. (c) scatter plot of permafrost C loss and soil T change during SAI implementation (2020-2069). Positive values indicate losses of soil C stocks to atmosphere.

line 129: what is meant by: this is slightly lower than the mitigating effects of G4 on permafrost soil C losses?

What we mean here is that the proportion of reduced CH₄ emissions (~40%) is slightly lower than the reduced soil carbon loss (~50%). The word “this” is a bit confusing, so we change it to “this ratio”.

Line 150-152: “Between 2020 and 2069 under G4, methane emissions are predicted to be about 40% lower than under RCP4.5, a difference significant at the 95% level for all ESMs (Table S1). **This ratio** is slightly lower than the mitigating effects of G4 on permafrost soil C losses (Table S1, Fig. 1)...”

line 131: how do you know that the reduction in microbial decomposition is the main driver for stabilizing permafrost C stock?

Changes in permafrost C stock can be simply expressed as inputs (from litter C and is proportional to NPP) minus outputs (soil heterotrophic respiration, which is directly proportional to current soil C pool and the decomposition rate). In Pinc-PanTher, CH₄ emissions are a constant fraction (2.3%; Schuur et al, 2013 and Schneider et al, 2015) of heterotrophic respiration from anoxic soils. Therefore, we believe that relatively low soil respiration under G4 is the main driver for stabilizing permafrost C stock. Furthermore, microbial decomposition is the main process of soil respiration in response to permafrost thaw (McCalley et al, 2014). Previously we used the term “microbial decomposition”, and we can change it to “soil respiration”.

Line 153-154: “and implies that the permafrost C stock is stabilized mainly by the reduced rate of **soil respiration** under the G4 scenario rather than changes in NPP caused by increased CO₂ fertilization or diffuse radiation.”

References:

- McCalley, C. K. et al. Methane dynamics regulated by microbial community response to permafrost thaw. *Nature* (2014). doi:10.1038/nature13798
- Schneider Von Deimling, T. et al. Observation-based modelling of permafrost carbon fluxes with accounting for deep carbon deposits and thermokarst activity. *Biogeosciences* (2015). doi:10.5194/bg-12-3469-2015
- Schuur, E. A. G. et al. Expert assessment of vulnerability of permafrost carbon to climate change. *Clim. Change* (2013). doi:10.1007/s10584-013-0730-7

lines 144/145: please explain why the models give different results.

We previously stated: “Table 1 shows that BNU and both MIROC models have different land surface models, soil depths and different ways of treating the snow cover, and these seem more likely to be drivers of changes in TSL and soil C losses.”

So part of the reason is the different land surface models. The other reason is likely to be the kinds of different Arctic responses analysed by Block et al (2019). Especially differences in land albedo feedbacks between the models.

We discuss these things in relation to an improved Table 1 and Uncertainties section.

Table 1. Characteristics of 7 ESMs, the TSL and NPP outputs of which are used to drive permafrost C dynamics.

ESMs (Ref)	Atmospheric models	Land models	Resolution (lon×lat) ^b	Maximum soil depth (layers)	Dynamic vegetation	Nitrogen cycle
BNU-ESM ²⁰	CAM3.5	CoLM	128 × 64	3.6 m (10)	Yes	No
CanESM2 ²¹	AGCM4	CLASS2.7	128 × 64	4.1 m (3)	Yes	No
HadGEM2-ES ²²	AOGCM	MOSES2.2	192 × 145	3.0 m (4)	Yes	No
GISS-E2-R ²³	GISS-ModelE	Model II-LS	144 × 90	3.3 m (6)	No	No
MIROC-ESM ²⁴	MIROC-AGCM	MATSIRO	128 × 64	14.0 m (6)	Yes	No
^a MIROC-ESM-CHEM ²⁵	MIROCAGCM&CHASER	MATSIRO	128 × 64	14.0 m (6)	Yes	No
NorESM1-M ²⁶	CAM4-Oslo	CLM4.0	144 × 96	42.1 m (15)	No	Yes

(Continued)

ESMs (Ref)	Multiple snow layers	Snow insulation effect	Latent heat of soil water	Differing frozen/unfrozen soil thermal conductivity	Snow thermal conductivity (W m ⁻¹ K ⁻¹)	Snow density (ρ) (kg m ⁻³)
BNU-ESM ²⁰	Yes	Yes	Yes	Yes	f(ρ ₂)	f(snow depth; compaction)
CanESM2 ²¹	No	Yes	Yes	Yes	f(ρ ₂)	exp(-time/t)
HadGEM2-ES ²²	No	No	Yes	Yes	Fixed at 0.265	Fixed at 250
GISS-E2-R ²³	Yes	Yes	Yes	Yes	Fixed at 0.3	Fixed at 330
MIROC-ESM ²⁴	Yes	Yes	Yes	No	Fixed at 0.3	Fixed at 300
^a MIROC-ESM-CHEM ²⁵	Yes	Yes	Yes	No	Fixed at 0.3	Fixed at 300
NorESM1-M ²⁶	Yes	Yes	Yes	Yes	f(ρ ₂)	f(snow depth; compaction)

a. MIROC-ESM-CHEM shares the common features as MIROC-ESM, but with an additional atmospheric-chemistry component (CHASER).

b. all outputs are downscaled to 0.25° × 0.25° resolution and bias-corrected based on observations and reanalysis data.

Line 159-216:

“Sources of Uncertainty in C estimates

The ESMs differ in very many ways and Table 1 lists only a few of them. Moore et al., (2020)³⁶ found that 4 of the same models as used here also produce a similar spread of differences (RCP4.5-G4) in mass balance for Greenland as found for C release from permafrost. This results from combinations of reasons: differences in the sensitivities of ESMs to greenhouse gas and SAI forcing means that simulated changes in AMOC, seasonal sea ice, cloud cover, specific humidity and hence longwave radiative absorption and accumulation rates all differ. A recent analysis⁴⁸ of ESM differences in the Arctic across 13 CMIP5 models including BNU-ESM, MIROC, NorESM1-M, CanESM2, GISS-E2-R, explains it largely in terms of differing estimates of surface albedo and Planck feedbacks which show the largest intermodel differences. Their analysis indicates that these differences arise not only from different degrees of simulated Arctic warming but also are partly related to the large differences in initial sea ice cover and surface temperatures⁴⁸. Fig. S6 shows the differences (G4-RCP4.5) between soil C annual losses as a function of differences in TSL and surface temperatures. There are marked differences between models, but it is clear that TSL is a far better predictor of C losses than surface temperature forcing, with TSL typically accounting for 30% of variance in C, whereas surface T reflects only 10% of the variability, and is not a significant predictor for 5 of the 7 ESMs (Fig. S6). Even though MIROC-ESM and MIROC-

ESM-CHEM share the same land surface module (Table 1), TSL accounts for about 20% of variability in C loss for MIROC-ESM, but twice as much with MIROC-ESM-CHEM. This spans almost the full range of sensitivity of C losses to changes in TSL. The remarkable differences in TSL change (G4-RCP4.5) between the two MIROC models may also be seen by comparing the monthly TSL difference (RCP4.5-G4) for all the ESMs (Fig. S7). This suggests that natural variability over the 50-year simulations is as large as differences due to model formulation, and hence supports the Block et al., (2019)⁴⁸ view that initial conditions for Arctic sea ice and phases of the long period climate cycles driving temperature play very important roles in explaining across ESM variability, and strengthens the utility of the multi-model ensemble mean as a way of neutralizing these.

In the case of permafrost thermal state, the lowest soil boundary is a critical uncertainty affecting the simulation of permafrost⁴⁹. Earlier generations of land surface model typically had a lower boundary at 3m soil depth, but in general the deeper the soil layers the better the model is at simulating the thermal condition. Modeled snow depth and surface and soil temperature offsets vary widely amongst models⁵⁰. Some have substantial biases in their soil temperature simulations (Fig. S8), that are attributed mainly to inappropriate description of the surface (vegetation, snow cover) and soil properties (soil texture, hydrology)⁵⁰. Snow cover plays an important role in modulating the variations of soil thermodynamics, and hence near-surface permafrost extent⁵¹. Several other factors such as differences in the treatment of soil organic matter, soil hydrology, surface energy calculations, model soil depth and vegetation also provide important controls on the simulated permafrost distribution⁵². Wang et al., (2016)⁵³ reviewed the performance of various land surface models, finding that models with better performance apply multilayer snow schemes. This allows them to simulate more realistic (stronger) insulation because they consider the snowpack's vertical structure and variability. They calculate the energy and mass balance in each snow layer, are able to capture nonlinear profiles of snow temperature and can also account for thermal insulation within the snowpack such as when the upper layer thermally insulates the lower layers⁵⁴. These models also incorporate storage and refreezing of liquid water within the snow and parameterise wet-snow metamorphism, snow compaction and snow thermal conductivity (Table 1), which have been found to be among the most important processes for good snow depth and surface soil temperature simulation⁵³. These factors all interact and obscure relationships, and we find that soil C losses did not exhibit statistically significant relationships with snow cover in the ESMs.

In comparison with the land surface models, differences in the treatment of stratospheric aerosol physics and chemistry⁴⁷ probably plays a minor role. For example, HadGEM2-ES adopts a stratospheric aerosol schemes to simulate the sulfate aerosol optical depth (AOD). BNU-ESM and MIROC-ESM use the prescribed meridional distribution of AOD recommended by the GeoMIP protocol⁵⁵. MIROC-ESM and MIROC-ESM-CHEM share most processes except for an interactive atmospheric chemistry module in MIROC-ESM-CHEM⁵⁶. But comparing the BNU-ESM results in Fig.1 with MIROC-ESM and MIROC-ESM-CHEM (which share similar SAI schemes), we see that

BNU-ESM produces much lower soil C losses than the other two which have very similar results. This is likely because BNU and both MIROC models have different land surface models, soil depths and different ways of treating the snow cover (Table 1). For the purposes of permafrost stability the general pattern of cooling produced seems dominant as soil temperatures appear the single most critical variable in determining the differences in response between G4 and RCP4.5. The patterns of cooling are well-replicated by all ESMs that have run G4³² (Fig. S2).

The lifespans of stratospheric sulfur aerosols in different regions and altitudes vary from a few months to several years⁵⁷. Since the G4 experiment does not remove CO₂ from the atmosphere, if radiative forcing by SAI were to suddenly cease, there would be a compensatory rapid global temperature rise, the "termination shock"⁵⁸. The rise in surface air temperatures following termination causes a strong air-soil temperature gradient that drives heat into the soil generating a sudden increase in permafrost emissions from 2070-2089 which partially offsets the carbon loses avoided during the G4 implementation (Fig. S9, Table S1). Hopefully, such an unplanned termination strategy would not be employed in reality⁵⁹."

Fig. S7. Monthly differences (G4-RCP4.5; units:°C) in TSL for each ESM over the 2020-2069 period.

line 148: isn't this result obvious if you force the PInc-PanTher model only with soil temperature? Or does this statement refer to the ESMs?

Soil temperature is the most critical variable for determining the differences between G4 and RCP4.5, but the regression line shows that it only accounts for 20-40% of the variance in soil C losses (Fig. S6). We also tried to force the standard PInc-PanTher model only with soil temperature, and found that the difference in soil C loss (RCP4.5 minus G4) increased. This difference is caused by the NPP increase under the G4 scenario being lower than RCP4.5 (Fig. S2).

Fig. S6. Scatterplots of the yearly (2020 to 2069) differences in soil C loss (G4-RCP4.5) as a function of surface temperature differences (top row) and TSL differences (bottom row). The R^2 and p values of the linear regressions are shown on each panel.

Fig. S2. Maps of soil T change between 2020 and 2069 under G4 and RCP4.5 scenarios, averaged from upper 3 meters. The relative soil T changes are $1.3 \pm 1.2^\circ\text{C}$ under G4, $2.2 \pm 1.2^\circ\text{C}$ under RCP4.5 and $-0.9 \pm 0.4^\circ\text{C}$ for G4-RCP4.5.

line 162: is deepest soil depth synonym with best soil model?

The model differences are addressed in the new Table 1 and Uncertainties section – we note it depends on many factors in addition to surface temperature forcing.

Previous authors have judged the pros and cons of each model through simple indicators such as maximum soil depth, the number of soil layers and snow layers, which are important indicators of the models' ability to simulate permafrost (Marchenko and Etzelmüller, 2013; Wang et al., 2016).

In fact, the soil temperature of the ESMs with deeper soil depth in Table 1 (NorESM1-M, MIROC-ESM and MIROC-ESM-CHEM) is closest to the reanalysis data (Fig. S8). However, the performance of NPP simulations shows another side (Fig. S3), as the

NorESM1-M without dynamic vegetation underestimates the NPP in the circum-Arctic region. Sometimes models are weighted by how well they reproduce historical observations (Olson and Evans, 2016), but a better initial simulation does not necessarily mean more accurate capture of future trends. Therefore, we choose to perform initial bias corrections and gave them equal weight, since each model in ensemble produces a possible future.

Line 181-198: “In the case of permafrost thermal state, the lowest soil boundary is a critical uncertainty affecting the simulation of permafrost⁴⁹. Earlier generations of land surface model typically had a lower boundary at 3m soil depth, but in general the deeper the soil layers the better the model is at simulating the thermal condition. Modeled snow depth and surface and soil temperature offsets vary widely amongst models⁵⁰. Some have substantial biases in their soil temperature simulations (Fig. S8), that are attributed mainly to inappropriate description of the surface (vegetation, snow cover) and soil properties (soil texture, hydrology)⁵⁰. Snow cover plays an important role in modulating the variations of soil thermodynamics, and hence near-surface permafrost extent⁵¹. Several other factors such as differences in the treatment of soil organic matter, soil hydrology, surface energy calculations, model soil depth and vegetation also provide important controls on the simulated permafrost distribution⁵². Wang et al., (2016)⁵³ reviewed the performance of various land surface models, finding that models with better performance apply multilayer snow schemes. This allows them to simulate more realistic (stronger) insulation because they consider the snowpack’s vertical structure and variability. They calculate the energy and mass balance in each snow layer, are able to capture nonlinear profiles of snow temperature and can also account for thermal insulation within the snowpack such as when the upper layer thermally insulates the lower layers⁵⁴. These models also incorporate storage and refreezing of liquid water within the snow and parameterise wet-snow metamorphism, snow compaction and snow thermal conductivity (Table 1), which have been found to be among the most important processes for good snow depth and surface soil temperature simulation⁵³. These factors all interact and obscure relationships, and we find that soil C losses did not exhibit statistically significant relationships with snow cover in the ESMs.”

References:

- Marchenko, S. & Etzelmüller, B. Permafrost: Formation and Distribution, Thermal and Mechanical Properties. in *Treatise on Geomorphology* (2013). doi:10.1016/B978-0-12-374739-6.00207-4
- Olson, R., Fan, Y. & Evans, J. P. A simple method for Bayesian model averaging of regional climate model projections: Application to southeast Australian temperatures. *Geophys. Res. Lett.* (2016). doi:10.1002/2016GL069704
- Wang, W. et al. Diagnostic and model dependent uncertainty of simulated Tibetan permafrost area. *Cryosphere* (2016). doi:10.5194/tc-10-287-2016

Minor comments and typos:

line 16: amount instead of amounts

line 27 and subsequent uses of ESM: ESM is normally used to as an abbreviation for Earth System Model, not Earth System Models. Thus, I suggest to write ESMs instead of ESM on line 41 and thereafter.

line 69: is close instead of are close

line 155: sentence seems incomplete

line 238: that only soil instead of that only on soil

line 239: change affects to affect

line 252: soil temperature is (add is)

Thanks for pointing out these issues. We revised the manuscript for all the minor comments and typos pointed out by the reviewer, and made remaining changes to enhance the grammar.

III. Response to Comments of Reviewer #3

Mitigation of Arctic permafrost carbon loss through stratospheric aerosol (SAI) geoengineering by Chen et al. uses data from GEO-MIP to in a permafrost carbon model to calculate avoided permafrost carbon feedback under stratospheric aerosol injection scheme. They use the avoided carbon emissions from permafrost in an integrated assessment model to calculate the economic impact of avoided carbon release.

The effects of SAI on permafrost have been studied before, but the economic analysis is novel to my knowledge. Use of several models to predict the warming pattern of SAI makes it possible to assess the related uncertainty better than in previous studies. The manuscript is mostly clearly written. I think the economic analysis is interesting and a welcome contribution to literature as an estimate. However, there are potentially either some issues with calculating the impacts of CH₄ emissions or clarification needed in describing methods.

We thank the Reviewer for sparing time to go through the manuscript, highlighting very important issues and providing valuable suggestions. We gave point-by-point response for each question and comment, and made more careful clarification in the manuscript. The content is as follows:

Major comments:

1) My main concern with the methodology of the manuscript is with making CH₄ emissions comparable with CO₂ emissions. The method description is somewhat unclear on this, so I don't know whether PAGE-ICE was using annual time-series of avoided CO₂ and CH₄ emissions or the aggregated CO₂ equivalent mean emission rate discussed in the main text.

The following comment assumes that they are using a standard GWPs to convert CH₄ to CO₂e. This approach have several problems (e.g Pierrehumbert, 2014, <https://doi.org/10.1146/annurev-earth-060313-054843>) that are not discussed or addressed in the manuscript.

In my opinion, a better approach would be use GWP or some related concept that consider change in emission rate rather than pulse emissions (<https://doi.org/10.1016/j.ijggc.2012.11.028>, <https://doi.org/10.1038/nclimate2998>, doi:10.1038/s41612-018-0026-8). Here the difference in CH₄ emissions between G4 and RCP4.5 seems to be rather constant.”*

Sorry for the unclear method description. We re-delineated the main process with reference to the PAGE technical manual.

(https://www.ibs.cam.ac.uk/fileadmin/user_upload/research/workingpapers/wp1104.pdf; page 16; this part is unchanged in PAGE-ICE).

First, permafrost C emissions under G4 and RCP4.5 are input to PAGE-ICE as the average annual emissions of CO₂ and CH₄.

Second, the permafrost fluxes are added to the anthropogenic global CO₂ and CH₄ emissions which follow the RCP4.5 scenario.

The concentration of CH₄ in the atmosphere is then estimated based on the lifetime of methane and cumulative emissions from anthropogenic and natural sources.

Finally, the radiative forcing from CH₄ in given time is calculated based on the concentration of CH₄. According to the following formula:

$$F_{CH_4}(t) = F_{CH_4}(t_0) + F_s \times (\sqrt{c(t)} - \sqrt{c(t_0)}) + over(N_2O)$$

where t_0 is the initial time (year 2015); c represents the concentration of CH₄; F_s is the forcing slope; and $over(N_2O)$ represents the overlap term between CH₄ and N₂O, which is typically under 0.1 W/m².

We add this text to the Methods section:

Line 341-343: “Here we added the permafrost annual CO₂ and CH₄ emissions we derived, to the RCP4.5 anthropogenic emissions projections which follow the SSP2 socio-economic scenario. Hence deriving estimates of the economic losses caused solely by PCF through the PAGE-ICE model.”

- *Here the difference in CH₄ emissions between G4 and RCP4.5 seems to be rather constant.*

The drier SAI climates might mean that CH₄ losses could be relatively less under G4 than under RCP4.5, but no one has good enough models to demonstrate that at present. So the relative differences between G4 and RCP4.5 might be larger than we suggest. We add a caveat to this effect:

Line 146-148: “Since climate under G4 would be drier than under RCP4.5^{36,47}, assuming the same proportionate release of CH₄ under both scenarios might tend to minimize (RCP4.5-G4) differences, and so be regarded as a conservative estimate.”

2) *The main policy-relevant point of the manuscript are the economic impacts of*

carbon released from the permafrost. To assess the importance of the findings, it would be good to know what are the economic impacts of avoided warming in G4 compared to RCP4.5. The PCF contributes further to this difference, but how much relatively? Also, some estimate on the relative contribution of avoided PCF to total avoided warming by SAI would be good.

An estimate on the relative contribution of avoided PCF to total avoided warming by SAI would be helpful. However, the analysis of the SAI economy focuses not only on the positive benefits of avoiding warming, but also on the possible negative effects of SAI implementation. The economic advantages of SAI are simply enormous, but the negative effects are very difficult to get at all accurate because of the "unknown unknowns" - that is unexpected surprises. Bickel and Agrawal (2013) suggested that even considering the probability and side effects of aborting geoengineering, SAI may still pass a cost-benefit test over a wide range of scenarios. Another approach has been to look at economic disparities under SAI compared with pure greenhouse gas forcing (Harding et al., 2020), finding significant benefits to SAI, though with very large uncertainties, and no analysis was done for G4 or RCP4.5.

These are some of the reasons why we have the statement "Instead of analyzing the cost-benefit of SAI as a whole, which is difficult because of unforeseen and presently unquantifiable damages from SAI"

Although no direct comparable results were found for G4, we might use the scenarios of the 2.5°C and 2°C targets (Yumashev et al, 2019) as an alternative, since it is difficult to achieve the 2°C goal with SAI alone (MacMartin and Kravitz, 2019).

So we used the PAGE-ICE model to assess the economic impacts of climate change, and found that under the RCP4.5, 2.5°C and 2°C targets, the economic losses due to climate change in 2069 will be 22, 10 and 6 trillion USD per year, respectively. But as emphasized we do not consider these estimates as at all reliable. But crudely, we consider 12~16 trillion USD per year as the economic benefit brought by G4's avoiding warming in 2069. The economic benefits by avoided PCF in 2069 would be about 0.55 trillion USD per year, thus the relative contribution of avoided PCF to total avoided warming by SAI would be about $1/20 \sim 1/30$.

New text:

Line 236-249: "An estimate of the relative contribution of avoided PCF to total avoided warming by SAI might be helpful. However, the analysis of the SAI economy should include not only on the positive benefits of avoiding warming, but also on the possible

negative effects of SAI implementation. The economic advantages of SAI are simply enormous, but the negative effects are very difficult to calculate because of the "unknown unknowns" - that is unexpected surprises. Bickel and Agrawal (2013)⁶⁵ suggested that even considering the probability and side effects of aborting geoengineering, SAI may still pass a cost-benefit test over a wide range of scenarios. Another approach has been to look at economic disparities under SAI compared with pure greenhouse gas forcing⁶⁶, although no analysis was done for G4 or RCP4.5, disparities are greatly reduced with SAI which is hence beneficial to social stability. Although analyzing the cost-benefit of SAI as a whole, is fraught with difficulty because of unforeseen and presently unquantifiable damages from SAI, here we used the PAGE-ICE model to evaluate the global economic impacts of PCF, and for reference total economic damage as well. We drive the PAGE-ICE model with permafrost CO₂ and CH₄ emissions under G4 and RCP4.5 scenarios (Methods). Under the Paris NDC commitments, the emission pathway is likely to resemble RCP4.5²⁷ and the socio-economic scenario be similar to IPCC SSP2."

Line 264-269: "For total economic loss estimates we used the PAGE-ICE model to assess the economic impacts of climate change, and found that under the RCP4.5, 2.5°C and 2°C targets⁶⁴, the economic losses due to climate change in 2069 will be US\$ 22, 10 and 6 trillion per year, respectively (Fig. S12). We consider these estimates simply as a comparative guide to see the PCF in context, and so crudely, we may consider US\$12~16 trillion per year as the economic benefit brought by G4's avoiding warming in 2069. The economic benefits by avoided PCF in 2069 would be about US\$ 0.6 trillion per year, thus the relative contribution of avoided PCF to total avoided warming by SAI would be about 1/20 ~ 1/30 of the total."

Fig. S12. Global economic impacts of climate change under RCP4.5, the 2.5°C and 2°C targets.

References:

Harding A.R., et al. Climate econometric models indicate solar geoengineering would reduce inter-country income inequality. Nat. Commun. (2020). doi:10.1038/s41467-019-13957-x

Bickel, J. E. & Agrawal, S. Reexamining the economics of aerosol geoengineering. *Clim. Change* (2013) doi:10.1007/s10584-012-0619-x.

Hinkel, J. et al. Coastal flood damage and adaptation costs under 21st century sea-level rise. *Proc. Natl. Acad. Sci. U. S. A.* (2014) doi:10.1073/pnas.1222469111.

MacMartin, D. G. & Kravitz, B. Mission-driven research for stratospheric aerosol geoengineering. *Proceedings of the National Academy of Sciences of the United States of America* (2019) doi:10.1073/pnas.1811022116.

Robock, A., Marquardt, A., Kravitz, B. & Stenchikov, G. Benefits, risks, and costs of stratospheric geoengineering. *Geophys. Res. Lett.* (2009) doi:10.1029/2009GL039209.

Yumashev, D. et al. Climate policy implications of nonlinear decline of Arctic land permafrost and other cryosphere elements. *Nat. Commun.* (2019). doi:10.1038/s41467-019-09863-x

Specific comments

Lines 22-23: Can you double-check this claim here. I didn't find the number 0.76 in the referenced article and six times larger than global average sounds surprisingly large.

Yes, we rounded 0.755 to 0.76 in previous manuscript.

According to the referenced article, Arctic region warmed at a rate of 0.755 °C per decade during 1998–2012, while global mean SAT increased at a rate of 0.112 °C per decade during this 'hiatus' period. So we say it is "more than 6 times". This reference captures well the accelerated warming in the Arctic and indicates that the amplified Arctic warming has contributed to the continual global warming, instead of a 'hiatus'.

Lines 36-67: Please, provide some references on moral hazard (decrease in motivation to decarbonise due to geoengineering).

We added some references on moral hazard in the manuscript,

Line 37-38: "Many have strong reservations about the moral implications of doing, or even researching, SAI if it reduces the motivation to decarbonize and emit less greenhouse gases (Robock 2008; The Royal Society 2009; Gunderson et al, 2019)."

References

Gunderson, R., Stuart, D. & Petersen, B. The Political Economy of Geoengineering as Plan B: Technological Rationality, Moral Hazard, and New Technology. *New Polit. Econ.* (2019). doi:10.1080/13563467.2018.1501356

Robock, A. 20 reasons why geoengineering may be a bad idea. *Bull. At. Sci.* (2008). doi:10.2968/064002006

The Royal Society. Geoengineering the climate: science, governance and uncertainty. *Clean Technologies and Environmental Policy* (2009). doi:10.1007/s10098-010-0287-3

Lines 42, 181 and 185: I think most countries have now NDCs instead of INDCs.

Yes, thanks for the correction. We have changed INDCs to NDCs in the manuscript.

Currently 186 countries have submitted their NDCs.

(<https://www4.unfccc.int/sites/NDCStaging/Pages/All.aspx>)

Line 42. I agree that RCP4.5 can arguably be labelled as most “policy-relevant”, but I would not say the same of G4. More realistic scenarios have slower ramp up of sulphur, in my opinion.

Possibly this could be so, but G4 is certainly policy-relevant because in practice SAI can only reduce global mean temperatures by up to 1-2°C since the effects of aerosol become strongly non-linear with increasing load as particles stick to each other becoming less radiatively active, and also fall out of the stratosphere faster (Niemeier & Timmreck, 2015). Additionally, larger aerosol loads run greater risk of unwanted side effects (Lawrence et al., 2018).

We modified in Line 46-54: “RCP4.5 is arguably the most policy-relevant scenario as the Nationally Determined Contributions (NDCs) greenhouse gas emissions framework would produce similar temperatures trajectories²⁷, and G4 is one of the most “realistic” GeoMIP experiments which uses RCP4.5 as a control run and models stratospheric aerosol injection rather than insolation reduction. Furthermore, the amount of aerosol specified could plausibly be delivered²⁸, and be relatively effective²⁹. The stratospheric sulfate loading required under G4 equates to about ¼ of the 1991 Pinatubo eruption per year, i.e. 5 Tg SO₂ per year into the equatorial lower stratosphere beginning in 2020 and ending in 2069 along with RCP4.5 greenhouse gas emissions³⁰. A more realistic scenario would probably include a ramp up of SAI, with various latitudes and altitudes of injection than in the simple G4 specification.”

Line 64: Explain please, what is the difference between NBP and NPP?

Net primary production (NPP) refers to the net production of organic carbon by plants in an ecosystem usually measured over a period of a year or more.

Net Biome Production (NBP) refers to the change in carbon stocks after episodic carbon losses due to natural or anthropogenic disturbances.

By definition, $NPP = NBP + \text{heterotrophic respiration} + \text{ecosystem disturbance}$.

[Redacted]

(schematic diagram from Kirschbaum et al, 2001)

Here, we intend to analyze the NBP output from the ESMs to show the relative contribution of Arctic permafrost to changes in global vegetation and soil carbon pools between RCP4.5 and G4 scenarios. But it turns out that this part is confusing and may make readers think that NBP is also an input parameter we use.

So we rewrite the analysis of NBP and NPP.

Line 84-90: “Most ESMs have lower NPP under G4 than RCP4.5 for the permafrost region because the cooling effect suppresses plant photosynthesis, the other models may have higher sensitivity to CO₂ fertilization than temperature, or the effects of diffuse light counteract the cooling effects (Fig. S3). But the net biological productivity (NBP; defined as NPP minus heterotrophic respiration and ecosystem disturbance) simulations of ESMs show small but positive differences for G4-RCP4.5 (Table S2), suggesting that, in contrast with the tropics, the slowing down of soil heterotrophic respiration due to cooling from SAI is the dominant response in the circum-Arctic permafrost region.”

References:

Kirschbaum, M. U. F., Eamus, D., Gifford, R. M., Roxburgh, S. H. & Sands, P. J. Definitions Of Some Ecological Terms Commonly Used In Carbon Accounting. in Net Ecosystem Exchange (2001).
<http://www.kirschbaum.id.au/definitions.pdf>

Lines 73-74: Can you make clear what is this range? Lowest and highest model results?

Yes, these are the maximum and minimum values from Plnc-Panther simulations driven by outputs of 7 ESMs. We modified this sentence as follow:

Line 64-67: Between 2020 and 2069, Plnc-Panther simulations of soil C, driven by outputs of 7 ESMs for the RCP4.5 projection, varied from 19.4 Pg C gain to 51.4 Pg C loss (mean 24.6 Pg C loss), while under G4 the ensemble mean was 11.3 Pg C loss (range: 28.7 Pg C gain to 43.7 Pg C loss).

Line 84: Usually, Scandinavia excludes Greenland and Finland. I guess that this is not intention here?

Yes agreed. We change it to “**Fennoscandia**”. We are including Finland but not Greenland since the permafrost is mostly covered by the ice sheet.

Line 102-104: “Losses under RCP4.5, and differences under G4 are most pronounced in Siberia, followed by Northern Canada and Alaska, while European Russia and **Fennoscandia** have least losses and differences.”

Line 90: Does this “pure greenhouse gas forcing” include other forcings of RCP4.5. This is a bit confusing.

Yes, it means RCP4.5 forcing which includes several other greenhouse gases and aerosols. What we mean here is a scenario not having SAI. This phrase is quite common in the literature. We can delete "pure", and the sentence stands since the weakened AMOC is simulated under many different scenarios in addition to RCP4.5

Line 108-110: “thus the mild climate conditions produced by the North Atlantic Drift in Northwestern Europe are maintained under SAI, but otherwise weakened under **greenhouse gas forcing**.”

Line 150: “Well-replicated” compared to what? Do you mean that models predict similar patterns?

In the new Fig. 1 and Fig. S2, we show that there are large differences in soil temperature rise across models, but the difference between RCP4.5 and G4 is relatively close, which indicates that the cooling effect of SAI is “well-replicated” among the models.

Fig. 1. (a) cumulative losses of circum-Arctic permafrost C storage from Plnc-PanTher between 2006 and 2100; each curve is the mean of two methods of calculation (Fig. S1; Methods). (b) mean annual temperature in the upper 3 m of soils, derived from bias-corrected ESM outputs (Table 1), solid curves are G4, dashed are RCP4.5. (c) scatter plot of permafrost C loss and soil T change during SAI implementation (2020-2069). Positive values indicate losses of soil C stocks to atmosphere.

Fig. S2. Maps of soil T change between 2020 and 2069 under G4 and RCP4.5 scenarios, averaged from upper 3 meters. The relative soil T changes are $1.3 \pm 1.2^\circ\text{C}$ under G4, $2.2 \pm 1.2^\circ\text{C}$ under RCP4.5 and $-0.9 \pm 0.4^\circ\text{C}$ for G4-RCP4.5.

Lines 136-156: The uncertainty discussion could be improved. First, the current text suggests more “Sources of Uncertainty in C estimates” than “Uncertainty in C estimates”. It could be useful to summarize the uncertainty range here and state what you exactly are discussing here, and then try to show where it stems from.

Yes, this is rewritten now:

Line 159-216:

“Sources of Uncertainty in C estimates

The ESMs differ in very many ways and Table 1 lists only a few of them. Moore et al., (2020)³⁶ found that 4 of the same models as used here also produce a similar spread of differences (RCP4.5-G4) in mass balance for Greenland as found for C release from permafrost. This results from combinations of reasons: differences in the sensitivities of ESMs to greenhouse gas and SAI forcing means that simulated changes in AMOC, seasonal sea ice, cloud cover, specific humidity and hence longwave radiative absorption and accumulation rates all differ. A recent analysis⁴⁸ of ESM differences in the Arctic across 13 CMIP5 models including BNU-ESM, MIROC, NorESM1-M, CanESM2, GISS-E2-R, explains it largely in terms of differing estimates of surface albedo and Planck feedbacks which show the largest intermodel differences. Their analysis indicates that these differences arise not only from different degrees of simulated Arctic warming but also are partly related to the large differences in initial sea ice cover and surface temperatures⁴⁸. Fig. S6 shows the differences (G4-RCP4.5) between soil C annual losses as a function of differences in TSL and surface temperatures. There are marked differences between models, but it is clear that TSL is a far better predictor of C losses than surface temperature forcing, with TSL typically accounting for 30% of variance in C, whereas surface T reflects only 10% of the variability, and is not a significant predictor for 5 of the 7 ESMs (Fig. S6). Even though MIROC-ESM and MIROC-ESM-CHEM share the same land surface module (Table 1), TSL accounts for about 20% of variability in C loss for MIROC-ESM, but twice as much with MIROC-ESM-CHEM. This spans almost the full range of sensitivity of C losses to changes in TSL. The remarkable differences in TSL change (G4-RCP4.5) between the two MIROC models may also be seen by comparing the monthly TSL difference (RCP4.5-G4) for all the ESMs (Fig. S7). This suggests that natural variability over the 50-year simulations is as large as differences due to model formulation, and hence supports the Block et al., (2019)⁴⁸ view that initial conditions for Arctic sea ice and phases of the long period climate cycles driving temperature play very important roles in explaining across ESM variability, and strengthens the utility of the multi-model ensemble mean as a way of neutralizing these.

In the case of permafrost thermal state, the lowest soil boundary is a critical uncertainty affecting the simulation of permafrost⁴⁹. Earlier generations of land surface model typically had a lower boundary at 3m soil depth, but in general the deeper the soil layers the better the model is at simulating the thermal condition. Modeled snow depth and surface and soil temperature offsets vary widely amongst models⁵⁰. Some have substantial biases in their soil temperature simulations (Fig. S8), that are attributed mainly to inappropriate description of the surface (vegetation, snow cover) and soil properties (soil texture, hydrology)⁵⁰. Snow cover plays an important role in modulating the variations of soil thermodynamics, and hence near-surface permafrost extent⁵¹. Several other factors such as differences in the treatment of soil organic matter, soil hydrology, surface energy calculations, model soil depth and vegetation also provide important controls on the simulated permafrost distribution⁵². Wang et al., (2016)⁵³ reviewed the performance of various land surface models, finding

that models with better performance apply multilayer snow schemes. This allows them to simulate more realistic (stronger) insulation because they consider the snowpack's vertical structure and variability. They calculate the energy and mass balance in each snow layer, are able to capture nonlinear profiles of snow temperature and can also account for thermal insulation within the snowpack such as when the upper layer thermally insulates the lower layers⁵⁴. These models also incorporate storage and refreezing of liquid water within the snow and parameterise wet-snow metamorphism, snow compaction and snow thermal conductivity (Table 1), which have been found to be among the most important processes for good snow depth and surface soil temperature simulation⁵³. These factors all interact and obscure relationships, and we find that soil C losses did not exhibit statistically significant relationships with snow cover in the ESMs.

In comparison with the land surface models, differences in the treatment of stratospheric aerosol physics and chemistry⁴⁷ probably plays a minor role. For example, HadGEM2-ES adopts a stratospheric aerosol schemes to simulate the sulfate aerosol optical depth (AOD). BNU-ESM and MIROC-ESM use the prescribed meridional distribution of AOD recommended by the GeoMIP protocol⁵⁵. MIROC-ESM and MIROC-ESM-CHEM share most processes except for an interactive atmospheric chemistry module in MIROC-ESM-CHEM⁵⁶. But comparing the BNU-ESM results in Fig.1 with MIROC-ESM and MIROC-ESM-CHEM (which share similar SAI schemes), we see that BNU-ESM produces much lower soil C losses than the other two which have very similar results. This is likely because BNU and both MIROC models have different land surface models, soil depths and different ways of treating the snow cover (Table 1). For the purposes of permafrost stability the general pattern of cooling produced seems dominant as soil temperatures appear the single most critical variable in determining the differences in response between G4 and RCP4.5. The patterns of cooling are well-replicated by all ESMs that have run G4³² (Fig. S2).

The lifespans of stratospheric sulfur aerosols in different regions and altitudes vary from a few months to several years⁵⁷. Since the G4 experiment does not remove CO₂ from the atmosphere, if radiative forcing by SAI were to suddenly cease, there would be a compensatory rapid global temperature rise, the "termination shock"⁵⁸. The rise in surface air temperatures following termination causes a strong air-soil temperature gradient that drives heat into the soil generating a sudden increase in permafrost emissions from 2070-2089 which partially offsets the carbon losses avoided during the G4 implementation (Fig. S9, Table S1). Hopefully, such an unplanned termination strategy would not be employed in reality⁵⁹."

Fig. S6. Scatterplots of the yearly (2020 to 2069) differences in soil C loss (G4-RCP4.5) as a function of surface temperature differences (top row) and TSL differences (bottom row). The R^2 and p values of the linear regressions are shown on each panel.

Table 1. Characteristics of 7 ESMs, the TSL and NPP outputs of which are used to drive permafrost C dynamics.

ESMs (Ref)	Atmospheric models	Land models	Resolution (lon×lat) ^b	Maximum soil depth (layers)	Dynamic vegetation	Nitrogen cycle
BNU-ESM ²⁰	CAM3.5	CoLM	128 × 64	3.6 m (10)	Yes	No
CanESM2 ²¹	AGCM4	CLASS2.7	128 × 64	4.1 m (3)	Yes	No
HadGEM2-ES ²²	AOGCM	MOSES2.2	192 × 145	3.0 m (4)	Yes	No
GISS-E2-R ²³	GISS-ModelE	Model II-LS	144 × 90	3.3 m (6)	No	No
MIROC-ESM ²⁴	MIROC-AGCM	MATSIRO	128 × 64	14.0 m (6)	Yes	No
^a MIROC-ESM-CHEM ²⁵	MIROCAGCM&CHASER	MATSIRO	128 × 64	14.0 m (6)	Yes	No
NorESM1-M ²⁶	CAM4-Oslo	CLM4.0	144 × 96	42.1 m (15)	No	Yes

(Continued)

ESMs (Ref)	Multiple snow layers	Snow insulation effect	Latent heat of soil water	Differing frozen/unfrozen soil thermal conductivity	Snow thermal conductivity (W m ⁻¹ K ⁻¹)	Snow density (ρ) (kg m ⁻³)
BNU-ESM ²⁰	Yes	Yes	Yes	Yes	f(ρ2)	f(snow depth; compaction)
CanESM2 ²¹	No	Yes	Yes	Yes	f(ρ2)	exp(-time/t)
HadGEM2-ES ²²	No	No	Yes	Yes	Fixed at 0.265	Fixed at 250
GISS-E2-R ²³	Yes	Yes	Yes	Yes	Fixed at 0.3	Fixed at 330
MIROC-ESM ²⁴	Yes	Yes	Yes	No	Fixed at 0.3	Fixed at 300
^a MIROC-ESM-CHEM ²⁵	Yes	Yes	Yes	No	Fixed at 0.3	Fixed at 300
NorESM1-M ²⁶	Yes	Yes	Yes	Yes	f(ρ2)	f(snow depth; compaction)

a. MIROC-ESM-CHEM shares the common features as MIROC-ESM, but with an additional atmospheric-chemistry component (CHASER).

b. all outputs are downscaled to 0.25° × 0.25° resolution and bias-corrected based on observations and reanalysis data.

Fig. S7. Monthly differences (G4-RCP4.5; units:°C) in TSL for each ESM over the 2020-2069 period.

If I understood correctly, the ESMs with deepest soils had least bias (Lines 260-261). Could you discuss, how the results would look, if these models were given more weight?

Previous authors have judged the pros and cons of each model through simple indicators such as maximum soil depth, the number of soil layers and snow layers, which are important indicators of the models' ability to simulate permafrost (Marchenko and Etzelmüller, 2013; Wang et al., 2016).

In fact, the soil temperature of the ESMs with deeper soil depth in Table 1 (NorESM1-M, MIROC-ESM and MIROC-ESM-CHEM) is closest to the reanalysis data (Fig. S8).

- how the results would look, if these models were given more weight?

These three models also show a more significant warming trend, so if weighted according to soil depth, the weighted ensemble C loss under RCP4.5 and G4 will be greater. However, the carbon loss avoided by G4 will not change much since scenario forcing dominates model differences (Table S1 and new Fig. 1)

For two reasons, we chose to use reanalysis data and MODIS products to bias-correct the outputs of 7 ESMs instead of weighting them based on historical performance.

Firstly, permafrost C dynamics depends on many factors in addition to surface temperature forcing. We can roughly judge the pros and cons of each model, but cannot give weights quantitatively. For example, NorESM1-M, which has the deepest soil depth, lacks dynamic vegetation (Table 1) and generates almost constant NPP simulations (Fig. S3).

Secondly, the initial bias is eliminated after bias correction, and what really matters is the future trend. Sometimes models are weighted by how well they reproduce historical observations (Olson and Evans, 2016), but a better initial simulation does not necessarily mean more accurate capture of future trends. Therefore, we choose to perform initial bias corrections and gave them equal weight, since each model in the ensemble produces a possible future.

References:

Marchenko, S. & Etzelmüller, B. Permafrost: Formation and Distribution, Thermal and Mechanical Properties. in *Treatise on Geomorphology* (2013). doi:10.1016/B978-0-12-374739-6.00207-4

Olson, R., Fan, Y. & Evans, J. P. A simple method for Bayesian model averaging of regional climate model projections: Application to southeast Australian temperatures. *Geophys. Res. Lett.* (2016). doi:10.1002/2016GL069704

Wang, W. et al. Diagnostic and model dependent uncertainty of simulated Tibetan permafrost area. *Cryosphere* (2016). doi:10.5194/tc-10-287-2016

Overall, this section feels like a list of model differences, but it's hard to understand the relevance of these differences on the main results of this study. I'm confused by the paragraph on ozone and NOx. The text says first that the interactive chemistry in MIROC-ESM-CHEM creates differences, but then MIROC-ESM and MIROC-ESM-CHEM are similar to each other and different from BNU-ESM. So the interactive chemistry doesn't make much of a difference? If so, could you be more explicit here for better readability? Does annual variability have any relevance here?

Agreed. For the purposes of permafrost stability the general pattern of cooling produced seems dominant as soil temperatures appear the single most critical variable in determining the differences in response between G4 and RCP4.5. The patterns of cooling are well-replicated by all ESMs that have run G4(Fig. S2).

We completely rewrote the Uncertainties section (Line 159-216, Given in a previous answer), and this paragraph was rewritten:

Line 199-209: "In comparison with the land surface models, differences in the treatment of stratospheric aerosol physics and chemistry⁴⁷ probably plays a minor role. For example, HadGEM2-ES adopts a stratospheric aerosol schemes to simulate the sulfate aerosol optical depth (AOD). BNU-ESM and MIROC-ESM use the prescribed meridional distribution of AOD recommended by the GeoMIP protocol⁵⁵. MIROC-ESM and MIROC-ESM-CHEM share most processes except for an interactive atmospheric chemistry module in MIROC-ESM-CHEM⁵⁶. But comparing the BNU-ESM results in Fig.1 with MIROC-ESM and MIROC-ESM-CHEM (which share similar SAI schemes), we see that BNU-ESM produces much lower soil C losses than the other two which have very similar results. This is likely because BNU and both MIROC models have different land surface models, soil depths and different ways of treating the snow cover (Table

1). For the purposes of permafrost stability the general pattern of cooling produced seems dominant as soil temperatures appear the single most critical variable in determining the differences in response between G4 and RCP4.5. The patterns of cooling are well-replicated by all ESMs that have run G4³² (Fig. S2)."

Line 155: What do you exactly mean by "diffusive effects"? Also "might be expected" sounds vague. Could you clarify the sentence?

We modify this sentence as follow:

Line 212-215: "The rise in surface air temperatures following termination causes a strong air-soil temperature gradient that drives heat into the soil generating a sudden increase in permafrost emissions from 2070-2089 which partially offsets the carbon losses avoided during the G4 implementation (Fig. S9, Table S1)."

Line 166: It seems that you are using methane's lifetime instead of its GPW value.

Yes, the lifetime of methane is taken into account, and the calculation method is explained in a previous answer.

L168: I think you mean CO2-equivalent instead of CO2-only (which could be understood to have only CO2 emissions)

We want to compare the reduced radiative forcing due to reduced CO₂ and CH₄ emissions, so the "CO₂-only flux" should be "CO₂ flux". In addition, this is only a simple estimation we made, not the calculation method of radiative forcing of CH₄ in the PAGE-ICE model. As mentioned before, in PAGE-ICE, the effects of permafrost CH₄ emissions are expressed by affecting CH₄ concentrations and then radiative forcing. The lifetime of methane is taken into account when calculating the remaining concentration in the atmosphere.

Line 173: I would suggest using the term carbon budget instead of "permissible anthropogenic C emission budget " and provide more references for it. I would also note that carbon release from permafrost is larger in RCP4.5 than in a scenario where global warming is limited under 2 degrees. Thus, your statement is somewhat exaggerating the amount of carbon released relative to 2-degree carbon budget.

Thanks to the reviewer for the valuable suggestion and comment.

We agree that the permafrost emissions under RCP4.5 are not suitable for direct comparison with the remaining carbon budget of the 2 °C target. Therefore, we refer

to the latest research and show the impact of permafrost emissions on the remaining carbon budget in a proportional form and the significance of the avoided PCF by G4.

According to a recent Nature article (Rogelj et al 2019) which synthesized recent estimates of remaining carbon budget, the world's remaining carbon budget for a 50% probability of staying within 2°C of warming is about 1400 Gt CO₂ (or 382 PgC; slightly lower than the estimates of Goodwin et al, 2018), while permafrost emissions could take an estimated 100 Gt CO₂ (or 27 PgC; lower than the estimates of Comyn-Platt et al, 2018) off this budget (and methane emissions are not included).

We revised in the manuscript as follow:

Line 225-227: "According to recent estimates of remaining carbon budget for maintaining a warming target of 2°C, carbon release from permafrost thaw could contribute about 10% of the world's budget⁵⁴⁻⁵⁶. Thus, the halved permafrost C emissions (mean 13 PgC) under G4 compared with RCP4.5 is a small, but significant fraction of the remaining carbon budget."

References:

Comyn-Platt, E. et al. Carbon budgets for 1.5 and 2°C targets lowered by natural wetland and permafrost feedbacks. Nat. Geosci. (2018). doi:10.1038/s41561-018-0174-9

Goodwin, P. et al. Pathways to 1.5 °c and 2 °c warming based on observational and geological constraints. Nat. Geosci. (2018). doi:10.1038/s41561-017-0054-8

Rogelj, J., Forster, P. M., Kriegler, E., Smith, C. J. & Séférian, R. Estimating and tracking the remaining carbon budget for stringent climate targets. Nature (2019) doi:10.1038/s41586-019-1368-z.

Lines 174-175: The sentence about temperatures feels a bit out of place here.

Yes, we removed this sentence.

Line 187: Could you combine the uncertainty from ESMs and from parametric uncertainty in PAGE-ICE? Now you have done them separately, but using the mean ESM temperature driver, Fig. 5 underestimates the total uncertainty.

Yes, we use CO₂ and CH₄ emissions (mean ± sd; in a Normal distribution) as the input to combine the uncertainty from permafrost emissions with uncertainty in PAGE-ICE. The new results show increased uncertainties (the 5-95% range), but the mean values have hardly changed.

Line 250-252: "Systematic uncertainties in the projected annual economic losses due to PCF are explored with uncertainty (mean ± 1σ, assuming a Gaussian distribution)

from permafrost emissions and a 100,000 member ensemble suite of simulations in Fig 5, (and the differences between RCP4.5 and G4 scenarios for each model in Fig. S11).”

Line 254-257: “The simulated annual economic losses and gains are very uncertain, with even about ¼ of the ensemble suggesting net economic benefits from the PCF under RCP4.5 and G4. This is due to GISS-E2-R and HadGEM2-ES simulating net C gains under the scenarios. The RCP4.5-G4 5-95% range differences are all positive with mean of US\$0.6 trillion and 5-95% range of 0.2–2.3 US\$ trillion in 2069 (Fig. 5)”.

New Fig. 5 and new Fig. S11:

Fig. 5. Predicted annual economic losses due to PCF under (a) G4 and (b) RCP4.5 scenarios, as well as (c) the difference (RCP4.5 - G4). Black lines represent the mean values, light and dark color intervals show 5–95% and 25–75% confidence intervals, respectively.

Fig. S11. The difference between RCP4.5 and G4 (RCP4.5 - G4) of predicted annual economic losses due to permafrost carbon–climate feedbacks. Black lines represent the mean values, light and dark color intervals show 5–95% and 25–75% confidence intervals, respectively.

Lines 214-215: The impact of prevention of permafrost carbon release is only a part of the full impact of SAI, and this sentence confuses that and the scope of this manuscript a bit. This would a good place to discuss how relevant PCF is in the big picture.

Thanks for the suggestion.

We deleted the sentence “Our study suggests that SAI provides a useful tool to buy extra time to cut emissions to reach a warming target of 2°C.”

We emphasize the difficulty in making full economic forecasts of SAI, but do so with caveats:

Line 236-246: “An estimate of the relative contribution of avoided PCF to total avoided warming by SAI might be helpful. However, the analysis of the SAI economy should include not only on the positive benefits of avoiding warming, but also on the possible negative effects of SAI implementation. The economic advantages of SAI are simply enormous, but the negative effects are very difficult to calculate because of the “unknown unknowns” - that is unexpected surprises. Bickel and Agrawal (2013)⁶⁵ suggested that even considering the probability and side effects of aborting geoengineering, SAI may still pass a cost-benefit test over a wide range of scenarios. Another approach has been to look at economic disparities under SAI compared with pure greenhouse gas forcing⁶⁶, although no analysis was done for G4 or RCP4.5, disparities are greatly reduced with SAI which is hence beneficial to social stability. Although analyzing the cost-benefit of SAI as a whole, is fraught with difficulty because of unforeseen and presently unquantifiable damages from SAI, here we used the PAGE-ICE model to evaluate the global economic impacts of PCF, and for reference total economic damage as well.”

Line 264-269: “For total economic loss estimates we used the PAGE-ICE model to assess the economic impacts of climate change, and found that under the RCP4.5, 2.5°C and 2°C targets⁶⁴, the economic losses due to climate change in 2069 will be US\$ 22, 10 and 6 trillion per year, respectively (Fig. S12). We consider these estimates simply as a comparative guide to see the PCF in context, and so crudely, we may consider US\$12~16 trillion per year as the economic benefit brought by G4's avoiding warming in 2069. The economic benefits by avoided PCF in 2069 would be about US\$ 0.6 trillion per year, thus the relative contribution of avoided PCF to total avoided warming by SAI would be about 1/20 ~ 1/30 of the total.”

Fig. S12. Global economic impacts of climate change under RCP4.5, the 2.5°C and 2°C targets.

Line 221: developed instead of proposed?

We modified the text accordingly.

Lines 269-271: Please, be more explicit what is exactly the input to PAGE-ICE. Does it take annual time-series of CO₂ and CO₂ from permafrost or some aggregated quantity?

Thanks for pointing it out. We modified it as suggested.

Line 341-343: “Here we added the permafrost annual CO₂ and CH₄ emissions we derived, to the RCP4.5 anthropogenic emissions projections which follow the SSP2 socio-economic scenario. Hence deriving estimates of the economic losses caused solely by PCF through the PAGE-ICE model”

Fig S6. Are the ranges given from ESMs? I’m a bit sceptical, how you derive 5% to 95% range with only seven models.

Yes, this boxplot is drawn based on the results driven by data from 7 ESMs. And sorry for the error in caption, the whiskers represent the maximum and minimum values, not the 5%-95% range. We calculated the 25%-75% range based on Excel's quartile function. In addition, we added a new plot with data points visible (Fig.S9).

Fig. S9. (a) Dot plot of integrated permafrost C loss. (b) box-plot of integrated permafrost C loss. Whiskers: minimum and maximum values; boxes: 25–75% range; horizontal lines: median; dots: ensemble mean.

Table S1. These are differences between G4 and RCP4.5, right? Can you specify that?

Yes, we revised the name of Table S1 to “Differences in PInc-PanTher estimates of permafrost soil C loss (units: Pg) and CH₄ emissions (units: Tg) between G4 and RCP4.5 (G4 minus RCP4.5).”

Technical corrections:

Several places: ESM is used incorrectly for plural form instead of ESMs.

Fig. S6. Somewhat relevant elsewhere too, but especially here the axes lines are much more clearly visible than the actual plot lines.

Thanks for pointing it out. We have checked the entire manuscript carefully with regard to grammar and typos. All line drawings have been redrawn for better visual effects.

Reviewers' comments:

Reviewer #2 (Remarks to the Author):

The authors did a great job in revising the manuscript. Thus I'm happy to accept the revised manuscript as is.

Ulrike Lohmann

Reviewer #3 (Remarks to the Author):

The authors have addressed my comments adequately and clarified and improved the manuscript. I recommend the manuscript for publication.

Reviewer #4 (Remarks to the Author):

This is a modelling study indicating that stratospheric aerosol injection would prevent a significant amount of permafrost carbon from being emitted to the atmosphere, under a realistic climate change mitigation scenario over the coming century. The scientific approach appears to be reasonable, on the whole, and the conclusions are interesting and important.

My role was to assess whether the authors have responded adequately to the comments of reviewer 1, however in reading the paper I have inevitably come up with some comments of my own.

In general, the response to the comments has been very thorough, and - indeed - satisfactory. The remaining queries/comments that I have are as follows:

Line 78 "indicating that scenario forcing dominates model differences." I'm afraid I just don't understand this wording, and the context doesn't seem to clarify it for me. Perhaps you could rephrase this.

Line 89 "in contrast with the tropics". This clause about the tropics suggests that the 'saved'/stabilised carbon in the tropics under G4 may be because NPP in the tropics is higher under G4 than RCP4.5 (which is not true of high latitudes). However, this could be stated more explicitly, and there is also no data in the manuscript on tropical NPP so this statement is not supported.

Line 135 'in degree' -> 'on degree'

Line 153 I find the statement (regarding methane) that "This ratio is slightly lower than the mitigating effects of G4 on permafrost soil C losses (Table S1, Fig. 1), and implies that the permafrost C stock is stabilized mainly by the reduced rate of soil respiration under the G4 scenario rather than changes in NPP caused by increased CO₂ fertilization or diffuse radiation.", a bit misleading.

The fractional change in permafrost carbon loss depends on the balance between NPP and respiration, and a 40% reduction in respiration (which the methane result suggest), could lead to a much larger fractional change in soil carbon emissions (if NPP does not change), since the net emissions are almost in balance, so a 40% reduction in respiration without a corresponding decrease in NPP could multiply the emissions many times over. Your statement implies that the 40% and 50% figures are somehow related to each other, which they essentially aren't. The results do imply that the NPP is also reduced under G4, so the statement is sort-of correct, but I would suggest reconsidering it.

Line 211-216, this is the only place where the termination is discussed and came as a surprise. It is mentioned in the introduction that the aerosol injection 'ends in 2069' but I found that was easy to miss. I would perhaps reword that part in the introduction (line 52) with a separate sentence or clause saying something like "aerosol injection is terminated in 2069 with immediate effect", just so it's easier to pick up on.

Line 226 Comyn-Platt et al is misquoted re: permafrost having a 10% impact on the carbon budget, in fact the 10% figure includes CO₂ from permafrost but methane from all wetlands globally. Instead of comparing your 50% reduction with these percentage impacts on carbon budget, it would be simpler (and equally valid) to just compare your figure of 13 PgC saved with the size of remaining carbon budget for e.g. reaching Paris agreement targets.

Line 257. Make it clearer that these figures are annual totals.

Line 312-314. Perhaps I am not understanding this correctly, but I am not convinced that the argument that the model with the most sophisticated snow scheme (and therefore - I guess you are assuming the most accurate soil temperatures?) giving carbon release close to the ensemble mean suggests that bias correction of soil temperature works. The carbon release is impacted significantly by NPP as well as temperature, therefore if the model with the best soil temperatures had an extreme NPP response it would not fall close to the ensemble mean, even if soil temperatures were all bias corrected perfectly. I suggest removing this sentence altogether as I don't think any evidence is really required that bias correction removes biases.

Line 324. Winter decomposition is not spurious, in fact there should be a lot of it and I am not convinced that this model is therefore correct/reasonable: see Natali et al (2019)

<https://www.nature.com/articles/s41558-019-0592-8>

Perhaps you could re-run without this constraint, or by simply reducing modelled respiration by a factor instead of setting it absolutely to zero (this is closer to what is observed in soils).

Units on Figure 4 are not consistent with the text (should be Tg), also should be clear if the units are in mass of carbon or methane (i.e. Tg C or Tg CH₄).

Figure S2: On the bottom row, the colour scale makes it difficult to distinguish which regions see higher temperatures, and which are lower. A colour scale where there is a sharper transition at zero (e.g. red-based colours above zero and blue-based ones below), would make this much easier to see.

I would also suggest that the Hope and Schaefer paper is referred to in discussion: values for economic impact of permafrost carbon emissions can be compared with those obtained in your study.

Response to reviewer comments on “Mitigation of Arctic permafrost carbon loss through stratospheric aerosol geoengineering.” by Chen et al.

Reviewers' comments:

Reviewer #2 (Remarks to the Author):

The authors did a great job in revising the manuscript. Thus I'm happy to accept the revised manuscript as is.

Ulrike Lohmann

Reviewer #3 (Remarks to the Author):

The authors have addressed my comments adequately and clarified and improved the manuscript. I recommend the manuscript for publication.

Response to Comments of Reviewer #4

Reviewer #4 (Remarks to the Author):

This is a modelling study indicating that stratospheric aerosol injection would prevent a significant amount of permafrost carbon from being emitted to the atmosphere, under a realistic climate change mitigation scenario over the coming century. The scientific approach appears to be reasonable, on the whole, and the conclusions are interesting and important.

My role was to assess whether the authors have responded adequately to the comments of reviewer 1, however in reading the paper I have inevitably come up with some comments of my own.

In general, the response to the comments has been very thorough, and - indeed - satisfactory.

We would like to thank the reviewer for the constructive comments and valuable suggestions, which helped us further improve the quality of the manuscript.

Below we address specific revisions in response to each comment. Line number references refer to those in the revised manuscript.

The remaining queries/comments that I have are as follows:

Line 78 "indicating that scenario forcing dominates model differences." I'm afraid I just don't understand this wording, and the context doesn't seem to clarify it for me. Perhaps you could rephrase this.

We modified Line 77-79: "Anomalies (G4-RCP4.5) exhibit much less scatter in permafrost C preserved (-13.7 ± 4.3 Pg) due to the well-replicated cooling effect of SAI (-0.9 ± 0.4 °C) among all ESMs that have run G4³² (Fig. S2), **indicating that despite model differences, SAI forcing produces consistent permafrost impacts.**"

Line 89 "in contrast with the tropics". This clause about the tropics suggests that the 'saved'/stabilised carbon in the tropics under G4 may be because NPP in the tropics is higher under G4 than RCP4.5 (which is not true of high latitudes). However, this could be stated more explicitly, and there is also no data in the manuscript on tropical NPP so this statement is not supported.

Xia et al. (2016; doi:10.5194/acp-16-1479-2016) suggests that stratospheric sulfate geoengineering will significantly increase plant photosynthesis rates in temperate and tropical regions, and reduce the photosynthesis rate in high latitudes. However, their conclusions are based on a single climate model and do not take into account the carbon-nitrogen cycle and dynamic vegetation.

Therefore, we decided to delete "in contrast with the tropics" to avoid unsupported conclusions. Moreover, tropical terrestrial carbon sinks are not very relevant to the subject of this manuscript.

Line 135 'in degree' -> 'on degree'

Thanks for the correction.

Line 153 I find the statement (regarding methane) that "This ratio is slightly lower than the mitigating effects of G4 on permafrost soil C losses (Table S1, Fig. 1), and implies that the permafrost C stock is stabilized mainly by the reduced rate of soil respiration under the G4 scenario rather than changes in NPP caused by increased CO2 fertilization or diffuse radiation.", a bit misleading.

The fractional change in permafrost carbon loss depends on the balance between NPP and respiration, and a 40% reduction in respiration (which the methane result suggest), could lead to a much larger fractional change in soil carbon emissions (if NPP does not change), since the net emissions are almost in balance, so a 40%

reduction in respiration without a corresponding decrease in NPP could multiply the emissions many times over. Your statement implies that the 40% and 50% figures are somehow related to each other, which they essentially aren't. The results do imply that the NPP is also reduced under G4, so the statement is sort-of correct, but I would suggest reconsidering it.

We agree that these two ratios are not suitable for direct comparison. We modified Line 153-154: "This ratio indicates that under the G4 scenario, the increase in soil respiration caused by PCF will be significantly suppressed by SAI."

Line 211-216, this is the only place where the termination is discussed and came as a surprise. It is mentioned in the introduction that the aerosol injection 'ends in 2069' but I found that was easy to miss. I would perhaps reword that part in the introduction (line 52) with a separate sentence or clause saying something like "aerosol injection is terminated in 2069 with immediate effect", just so it's easier to pick up on.

As suggested, we modified in the introduction: "The stratospheric sulfate loading required under G4 equates to about ¼ of the 1991 Pinatubo eruption per year, i.e. 5 Tg SO₂ per year into the equatorial lower stratosphere beginning in 2020 along with RCP4.5 greenhouse gas emissions³⁰. **Aerosol injection will end in 2069 under G4, followed by compensatory rapid warming¹⁴.** A more realistic scenario would probably include a ramp up **and down** of SAI, with various latitudes and altitudes of injection than in the simple G4 specification."

Line 226 Comyn-Platt et al is misquoted re: permafrost having a 10% impact on the carbon budget, in fact the 10% figure includes CO₂ from permafrost but methane from all wetlands globally. Instead of comparing your 50% reduction with these percentage impacts on carbon budget, it would be simpler (and equally valid) to just compare your figure of 13 PgC saved with the size of remaining carbon budget for e.g. reaching Paris agreement targets.

We modified Line 225-227: "The remaining global carbon budget for reaching the 2°C warming target is about 400 PgC^{60,61}, but the emitted carbon from thawing permafrost could reduce it by about 30 PgC^{61,62}. Thus the 6-19 (mean 14) PgC not released by permafrost under G4 compared with RCP4.5 is a small, but significant fraction of the remaining emission budget."

Line 257. Make it clearer that these figures are annual totals.

We modified this: “The RCP4.5-G4 5-95% range differences are all positive with mean of US\$0.6 trillion/yr and 5-95% range of US\$ 0.2–2.4 trillion/yr in 2069 (Fig. 5).”

Line 312-314. Perhaps I am not understanding this correctly, but I am not convinced that the argument that the model with the most sophisticated snow scheme (and therefore - I guess you are assuming the most accurate soil temperatures?) giving carbon release close to the ensemble mean suggests that bias correction of soil temperature works. The carbon release is impacted significantly by NPP as well as temperature, therefore if the model with the best soil temperatures had an extreme NPP response it would not fall close to the ensemble mean, even if soil temperatures were all bias corrected perfectly. I suggest removing this sentence altogether as I don't think any evidence is really required that bias correction removes biases.

We removed this sentence as suggested.

Line 324. Winter decomposition is not spurious, in fact there should be a lot of it and I am not convinced that this model is therefore correct/reasonable: see Natali et al (2019) <https://www.nature.com/articles/s41558-019-0592-8>

Perhaps you could re-run without this constraint, or by simply reducing modelled respiration by a factor instead of setting it absolutely to zero (this is closer to what is observed in soils).

We agree that decomposition in winter does proceed in the upper layer of soils, thus the word “spurious” is incorrect. We modified Line 323-326: “The standard PInc-PanTher uses a truncated Q_{10} function, which assumes zero soil respiration when soil temperature is at, or below, the freezing point⁸. However, decomposition in winter does proceed in the upper layer of soils, so we assume that the winter respiration rates of the upper layer increase by a factor of 2.9 per 10°C soil temperature⁷³.”

As suggested, we re-simulated permafrost soil C decomposition below zero and set Q_{10} in winter to 2.9 in the upper layer according to Natali et al (2019). We updated all the results (please see the new manuscript) and found that they were not significantly different from the previous ones, so our main conclusions remain the same.

Units on Figure 4 are not consistent with the text (should be Tg), also should be clear if the units are in mass of carbon or methane (i.e. Tg C or Tg CH₄).

We updated Figure 4 as suggested.

Figure S2: On the bottom row, the colour scale makes it difficult to distinguish which regions see higher temperatures, and which are lower. A colour scale where there is a sharper transition at zero (e.g. red-based colours above zero and blue-based ones below), would make this much easier to see.

As suggested, we used a new colour bar and redrew Fig.2, Fig.S2, S4 and S5 for better visual performance. Please see the new manuscript.

I would also suggest that the Hope and Schaefer paper is referred to in discussion: values for economic impact of permafrost carbon emissions can be compared with those obtained in your study.

As suggested, we added the comparison in discussion: “Hope and Schaefer (2016)⁶⁴ suggested that under the medium emission scenario (A1B) and aggressive abatement policy (2015r5low), the total economic impacts of carbon emissions from thawing permafrost by **2200** can reach US\$43 trillion and 20 trillion, respectively.”